# Molecular dissection of the soluble photosynthetic antenna from the cryptophyte alga *Hemiselmis andersenii*

Harry W. Rathbone [1,2,7], Alistair J. Laos[3], Katharine A. Michie [1,2,4], Hasti Iranmanesh[1,2], Joanna Biazik[4], Sophia C. Goodchild[5], Pall Thordarson [3], Beverley R. Green [6] & Paul M. G. Curmi [1,2✉]

Cryptophyte algae have a unique phycobiliprotein light-harvesting antenna that fills a spectral gap in chlorophyll absorption from photosystems. However, it is unclear how the antenna transfers energy efficiently to these photosystems. We show that the cryptophyte *Hemiselmis andersenii* expresses an energetically complex antenna comprising three distinct spectrotypes of phycobiliprotein, each composed of two αβ protomers but with different quaternary structures arising from a diverse α subunit family. We report crystal structures of the major phycobiliprotein from each spectrotype. Two-thirds of the antenna consists of *open* quaternary form phycobiliproteins acting as primary photon acceptors. These are supplemented by a newly discovered *open-braced* form (~15%), where an insertion in the α subunit produces ~10 nm absorbance red-shift. The final components (~15%) are *closed* forms with a long wavelength spectral feature due to substitution of a single chromophore. This chromophore is present on only one β subunit where asymmetry is dictated by the corresponding α subunit. This chromophore creates spectral overlap with chlorophyll, thus bridging the energetic gap between the phycobiliprotein antenna and the photosystems. We propose that the macro-molecular organization of the cryptophyte antenna consists of bulk *open* and *open-braced* forms that transfer excitations to photosystems via this bridging *closed* form phycobiliprotein.

[1] School of Physics, The University of New South Wales, Sydney, NSW 2052, Australia. [2] School of Biotechnology and Biomolecular Sciences, The University of New South Wales, Sydney, NSW 2052, Australia. [3] UNSW RNA Institute and School of Chemistry, The University of New South Wales, Sydney, NSW 2052, Australia. [4] Mark Wainwright Analytical Centre, The University of New South Wales, Sydney, NSW 2052, Australia. [5] School of Molecular Sciences, Macquarie University, Sydney, NSW 2109, Australia. [6] Department of Botany, University of British Columbia, Vancouver, BC, Canada. [7] Present address: UMR144 Cell Biology and Cancer, Institut Curie, Paris 75005, France. ✉email: p.curmi@unsw.edu.au

Cryptophytes are single-celled algae that obtained their chloroplasts by secondary endosymbiosis of a red alga[1–3]. The ancestral red algal light harvesting antenna, the phycobilisome, is a 1.2–20 MDa complex attached to the stromal side of the thylakoid membranes which creates an energetic funnel from the outer reaches of the complex down to the integral membrane photosystems[4,5]. In contrast, the cryptophyte antenna is comprised of repurposed phycobilisome components[3] densely-packed between the photosynthetic membranes in the thylakoid lumen.

Each cryptophyte phycobiliprotein (PBP) is a (hetero)dimer of two αβ protomers where each protomer contains a unique α subunit and a common β subunit, where the latter has descended virtually unchanged from the red algal phycobilisome phycoery-thrin (PE) β protein[6]. Crystal structures show that the crypto-phyte PBPs take on two distinct quaternary structures: the *closed* form[7,8] and the *open* form[9]. The latter form results from a single amino acid insertion in the α subunit which alters the packing between αβ protomers, creating a large solvent filled cavity[9]. Sequence data shows that *open* forms exist only in the *Hemiselmis* genus[10].

Since the discovery of the cryptophyte light harvesting system[11–13], it has been widely believed that the soluble antenna is comprised largely of copies of a single PBP per organism with a specific absorption maximum between 545 and 645 nm giving each PBP its name (e.g. PE545)[1,14–16]. This is despite evidence to the contrary coming from absorption spectroscopy and isoelectric focusing[11–13,17–22] as well as evidence of multiple nuclear α subunit genes from the genome sequence of *Guillardia theta* plus transcriptomic data[17,23]. In fact, all 20 α subunit genes in *G. theta* are expressed in culture[24]. Finally, for many cryptophytes there appears to be an impassable spectral gap between the fluorescence emission of the major PBPs and the 670–710 nm absorption maxima of the membrane-bound photosystems.

In this paper we demonstrate that the cryptophyte *Hemiselmis andersenii*[25] has an energetically complex antenna with multiple protein components possessing different spectral and structural characteristics, along with a new quaternary structure. Using energetic considerations, we define specific roles for each of these proteins in order to generate a functional antenna and formulate a simple model for the organization of this antenna in vivo. Finally, we show that this model appears to be general, harkening back to previous studies on other cryptophytes with unexplained spectral features.

## Results and discussion

### *Hemiselmis andersenii* soluble antenna is composed of multiple PBPs that fall into three spectrotypes.

Analysis of the tran-scriptomes of four strains of *H. andersenii* yielded 22 unique cryptophyte α subunit sequences (Fig. 1a; Supplementary Table 1; Methods)[17,18]. The final mature sequence set consists of eight *closed* forms ($Ha\alpha^C_1$ to $Ha\alpha^C_8$, Fig. 1a) and eleven *open* forms ($Ha\alpha^O_1$ to $Ha\alpha^O_{11}$) (Fig. 1a). Three additional *open* forms contain a seven-residue insertion between the second stand of the β ribbon (*S2*) and the α helix (*H1*) and these three sequences are here termed *open-braced* ($Ha\alpha^{OB}_1$ to $Ha\alpha^{OB}_3$; Fig. 1a). Two *open* forms, $Ha\alpha^O_3$ and $Ha\alpha^O_4$, had smaller insertions in this region. The eight *closed* forms fit largely into distinct $\alpha_L$ (long; $Ha\alpha^C_1$ and $Ha\alpha^C_5$) and $\alpha_S$ (short; $Ha\alpha^C_2$, $Ha\alpha^C_3$, $Ha\alpha^C_4$ and $Ha\alpha^C_6$) categories[10]. Thus, after the emergence of *open* forms in the *Hemiselmis* genus[9], closed forms still remain suggesting a func-tional reason.

The soluble *H. andersenii* PBPs were extracted and first partitioned into two distinct colored fractions, one pink, the other purple, by anion exchange chromatography (Fig. 1b left and right,

respectively; Supplementary Fig. 1; Methods). Each of these two fractions was further separated into a number of discrete protein fractions by cation exchange chromatography, that are labelled according to their distinguishing wavelength maximum and a letter indicating its relative abundance (Fig. 1b).

Linear absorption spectra show that each isolate fell into one of three distinct categories (Fig. 1c; Supplementary Note 1). The major spectrotype, *Ha*PE555, has an absorption peak at 551 ± 2 nm (Fig. 1c top row); the second, *Ha*PE560, has an absorption peak at 562 ± 2 nm (Fig. 1c middle row); and the last, *Ha*PE645, has a peak at 562 ± 2 nm with a smaller secondary peak at 645 ± 2 nm (Fig. 1c bottom row).

Peptide identification by fragment LC-MS/MS and intact mass spectrometry were combined to match sequence data with purified protein fractions. Nine α subunit sequences were identified with high confidence by their MASCOT score (Fig. 1d; Methods). Using intact mass spectrometry, the most probable pairings to produce mature $\alpha_1\beta.\alpha_2\beta$ PBPs were: $Ha\alpha^O_1$—$Ha\alpha^O_2$ (fraction 555 A), $Ha\alpha^{OB}_1$—$Ha\alpha^{OB}_1$ (560 A), $Ha\alpha^C_1$—$Ha\alpha^C_2$ (645 A and 645B), $Ha\alpha^C_1$—$Ha\alpha^C_3$ (645B and 645 C), and possibly $Ha\alpha^O_3$—$Ha\alpha^O_4$ (XA) and $Ha\alpha^C_1$—$Ha\alpha^C_4$ (645 C) (Fig. 1e; Methods).

Abundances for each light harvesting protein were estimated by comparing peak areas in chromatograms (Methods). Each spectro-type appears to have one clear majority fraction. The largest peak of *Ha*PE555 (555 A) is by far the most abundant PBP constituting approximately 63% of the overall protein complement (with other *Ha*PE555 spectrotype peaks contributing 8%). The *Ha*PE560 peak constitutes 14% of the protein complement and the largest peak of *Ha*PE645 constitutes 8% (with other *Ha*PE645 spectrotype pro-teins contributing 7%; Supplementary Table 2). This suggests that the spectrotypes are approximately in the ratio of 5:1:1 (*Ha*P-E555:*Ha*PE560:*Ha*PE645). These data show that there is an overwhelming majority of a single PBP in the antenna of *H. andersenii*, that of spectrotype *Ha*PE555, explaining why the one-PBP-per-organism convention has persisted.

Fluorescence excitation-emission maps were produced for each spectrotype to provide clues to the overall energetic architecture of the antenna (Fig. 1f; Methods). These maps show that *Ha*PE560 and *Ha*PE555 spectrotypes have unusually narrow spectra[9] with similar excitation and emission wavelength ranges (Fig. 1f; *Ha*PE555: excitation peak 551 ± 2 nm, emission peak 572 ± 1 nm; *Ha*PE560: excitation peak 562 ± 2 nm, emission peak 576 ± 1 nm). *Ha*PE645 has many of the same features as the other two spectrotypes (excitation peak 562 ± 2 nm and major emission peak 572 ± 1 nm), however, it has an extra peak absorbing at 646 ± 2 nm and emitting at 650 ± 1 nm plus a cross-peak connecting the two wavelength regions (excitation at 562 ± 2 nm, emission at 650 ± 1 nm). The excitation-emission map for *Ha*PE645 suggests that this protein acts as an energy adaptor with excitation transfer from higher energy (562 nm) to lower energy (650 nm) chromophores within the protein.

### Crystal structures of *Ha*PE555.

Four new structures of *Ha*PE555 were determined (Supplementary Fig. 2a; Table 1; resolutions: 1.57 Å, 1.67 Å, 1.73 Å and 1.95 Å). Three of these (PDB: 8EL3, 8EL4, 8EL5) were derived from peak 555 A (Fig. 1b left) and the remaining one (PDB: 8EL6) was derived from peak 555B (Fig. 1b). They all have the *open* form quaternary structure with two distinct, but nearly identical α subunits (Supplementary Fig. 2a bottom). Each protein in the new structures matches that of 4LMX to RMSD 0.46 ± 0.17 Å over 2937 ± 124 atoms, however, they differ in their packing and a small alteration in secondary structure in the structure from chromatogram peak 555B (8EL6), (Supplementary Note 2; Supplementary Fig. 3).

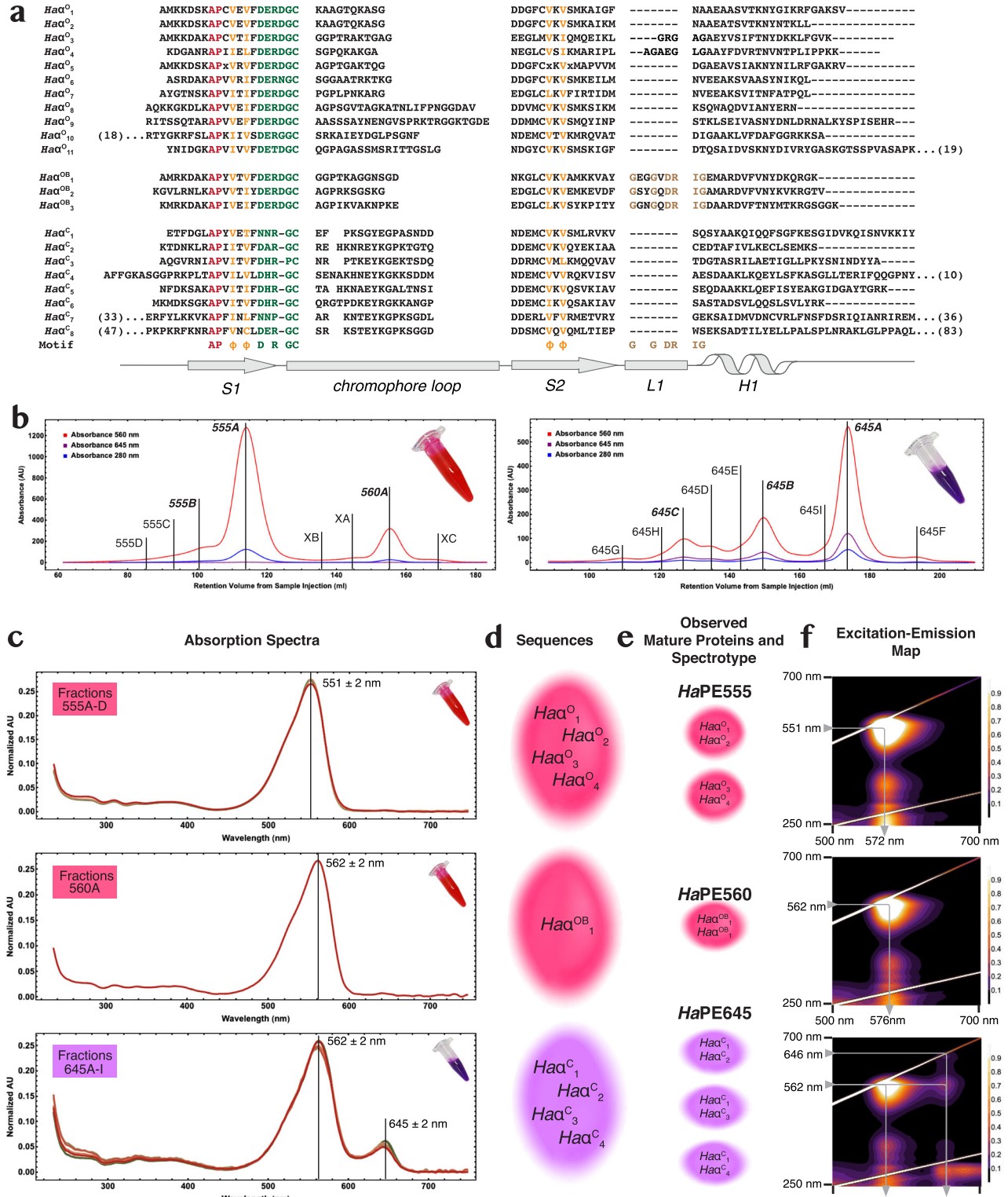

All five crystal forms of protein *Ha*PE555 are constructed from continuous filaments of PBPs, which are organized into two-dimensional rafts (Supplementary Fig. 2b; Supplementary Note 3). The filaments are generated via knobs-into-holes packing but differ in the distances between adjacent molecules along the filament (having either tight or loose interfaces; Supplementary Fig. 2b; Supplementary Table 3). This packing brings the chromophores of neighboring PBPs closer together increasing the possibility of energy transfer (Supplementary Fig. 2c–e). This

filament arrangement appears to be unique to *Ha*PE555, although filaments have been observed for the *closed* forms: *Cp*PE566 from *Cryptomonas pyrenoidifera*[10]; and *Ps*PE545 from *Proteomonas sulcata*[26].

**Crystal structure of *Ha*PE560.** The crystal structure of *Ha*PE560 was determined at 1.45 Å resolution (PDB 7SSF, Fig. 2a; Table 1). It shows a symmetric $(\alpha\beta)_2$ dimer (Fig. 1e middle; Fig. 2a) that resembles the *open* form structure of *Ha*PE555 with which it

**Fig. 1 *Hemiselmis andersenii* soluble antenna is composed of multiple PBPs that fall into three spectrotypes. a** Alignment of all *Hemiselmis andersenii* α subunit transcripts, clustered by *open*, *open-braced* and *closed* state predicted quaternary structures. The structural motif is at the bottom. Color coding: red —identical; orange—conserved property; green—chromophore site; and brown—*open-braced* form *L1* motif. Numbers in parentheses indicate additional N- and C-terminal residues. **b** Cation exchange chromatograms show the zoo of light harvesting proteins as peaks in chromatograms. Two fractions: pink and purple (see images inset in chromatograms) were first separated by anion exchange chromatography. The left chromatogram is for the pink fraction while the right chromatogram is for the purple fraction. The largest peaks are shown in bold with all peaks labelled by their spectrotype. The chromatograms show absorbance at three visible wavelengths: 560 nm red; 645 nm purple; and 280 nm blue. **c**–**f** Diagram showing the flow of characterization for peaks isolated in **b** from: **c** absorption spectra of peaks to **d**. identification of α subunit components via fragment LC-MS/MS to **e**. identification of purified mature proteins via intact mass spectrometry to **f**. characterization of spectrotypes by fluorescence emission-excitation maps. **c** Overlays of visible absorption spectra for each chromatographically separated species belonging to the three principal spectrotypes. The *Ha*PE555 (top) and *Ha*PE645 (bottom) panels contain an overlay of multiple spectra from chromatographically distinct peaks, while the *Ha*PE560 (center) panel contains a single spectrum. **d** Transcript sequences that have been identified via fragment LC-MS/MS to correspond to peaks in the chromatograms sorted by spectrotype. **e** These sequences are assembled into mature proteins where the component α subunits are paired based on the highest scoring observations by intact mass spectrometry. **f** Excitation-emission maps for the most abundant protein from each of the three spectrotypes. The arrows map the energy transfer pathways through each protein. The vertical axis is the excitation wavelength and the horizontal axis is the emission wavelength. The highest peaks have been truncated so as to highlight the smaller features in the heat maps.

shares the same chromophore arrangement despite the spectral differences (Supplementary Table 4; Fig. 1c). However, the *Ha*PE560 structure shows several critical differences. The key structural difference between *Ha*PE560 and *Ha*PE555 is that the *Ha*PE560 α subunit contains an inserted loop of seven residues (*L1*) between β-strand *S2* and α-helix *H1* (Fig. 1a). This *L1* loop partially closes the entrance to the central solvent filled cavity of *Ha*PE560 when compared to *open* form PBPs such as *Ha*PE555 (Fig. 2a–c; Supplementary Fig. 4a). In the *open* form, this cavity is created by the insertion of an Asp residue just prior to the chromophore binding Cys residue[9] where this change in quaternary structure reduces the buried solvent accessible surface area between two αβ protomers to $1084 \pm 63$ Å$^2$ for *Ha*PE555 compared to 2191 Å$^2$ for the *closed* form of *Chroomonas Cs*PC645. In *Ha*PE560, the insertion of the *L1* loop redresses this loss of interaction by increasing the buried surface area between two αβ protomers to $1452 \pm 2$ Å$^2$. Thus, the evolution of the *L1* loop strengthens and stabilizes the interface between adjacent α subunits connecting the two αβ protomers, thereby consolidating the *open* form quaternary structure. Given this, we call the new quaternary structure of *Ha*PE560 the *open-braced* form.

The *L1* loop insertion alters the conformation and geometric arrangement of the α chromophore in the *open-braced* quaternary structure. The *L1* loop forms a twisted β hairpin and scaffolds the opposite α chromophore (Fig. 2a–c; Supplementary Fig. 5a). The propionate group emanating from pyrrole ring B forms a hydrogen bond to the backbone nitrogen of Gly51 forming the tip of the *L1* hairpin (tight-turn), stabilizing the hairpin structure (Fig. 2d). The side chain of Asp53 on the descending strand of the *L1* β hairpin contacts the opposite α chromophore and forms a hydrogen bond to the nitrogen of pyrrole ring A (Fig. 2d). This interaction with Asp53 pulls pyrrole ring A away from the usual *open* form conformation and rotates pyrrole ring A (Fig. 2e; Supplementary Fig. 6a–c). It is therefore likely responsible for the ~10 nm wavelength shift of *Ha*PE560 absorption compared to *Ha*PE555 (which has identical chromophore placements) (Fig. 2b–e; Supplementary Fig. 6a–c).

Examination of transcriptome data for *Hemiselmis tepida* and *Hemiselmis rufescens* shows that each of these organisms contains α subunit sequences that show a homologous *L1* loop insertion that is a signature for the *open-braced* form (Fig. 2f; Supplementary Note 4). The transcriptomes for these organisms also contain multiple *open* form α subunit sequences, outnumbering the *open-braced* form (Supplementary Fig. 7; *open*:*open-braced* transcripts – 11:3, 8:6 and 8:1 for *H. andersenii*, *H. refescens* and *H. tepida*, respectively). Thus, the more stable *open-braced* form has not displaced the *open* form, suggesting there is a functional reason for maintaining these two forms.

**Crystal structure of *Ha*PE645.** The structure of the *closed* form *Ha*PE645 provides an explanation for its unusual spectral properties. The crystal structure of the major *Ha*PE645 species (Fig. 1b) was determined at 1.49 Å resolution (PDB 7SUT; Fig. 3a; Table 1) and shows a typical *closed* form PBP[7–9] with α$_L$ and α$_S$ sequences corresponding to *Ha*α$^C_1$ and *Ha*α$^C_2$, respectively (Fig. 1a). Our electron density maps clearly demonstrate that the chromophore attached to Cys82 on the β subunit in the α$_S$β protomer is unambiguously PCB (which causes the secondary absorption peak for this protein at 645 nm; Fig. 3b; Supplementary Fig. 8), while the chromophore attached to Cys82 on the β subunit in the α$_L$β protomer is unambiguously PEB (Fig. 3c; Supplementary Fig. 8). Such an asymmetry has not been observed previously for cryptophyte β subunits within a single PBP and poses an interesting question— how can two identical β-subunits within the same organism acquire different chromophores at position 82?

The N-terminal region of the α subunit appears to dictate the chemical nature of the chromophore attached to Cys82 on the associated β subunit. In each αβ protomer, the N-terminal tail of the respective α subunit interacts with pyrrole ring D of the β82 chromophore. For the α$_S$β protomer, the side chain of Leu6 packs against the face of pyrrole ring D of PCB β82 (Fig. 3d). This pocket is specific for PCB and incompatible with a PEB chromophore due to the steric clash that would be created by Leu6 (Fig. 3d compared to Fig. 3f). There is a similarity between *Ha*PE645 and *Cs*PC645 when looking at the residues surrounding the β82 PCB chromophore (compare Fig. 3d, e, respectively). The conformation of PCB β82 is nearly identical when comparing the α$_S$β protomer of *Ha*PE645 to both β82 sites in *Cs*PC645[9]. At all three sites, leucine makes an equivalent interaction with PCB β82 pyrrole ring D (Fig. 3d, e). In other cryptophyte-PC forms (*Hv*PC612, *Cs*PC630 and *Hp*PC577), this residue is replaced by a glutamic acid or serine which induces a rotation of pyrrole ring D of the chromophore, tuning the absorption away from 645 nm[9,10] (Fig. 3g; Supplementary Fig. 6d, e; Supplementary Note 5).

How can two identical β-subunits acquire different chromophores at position 82? Lyase enzymes mediate the stereospecific ligation of the correct chromophore to a specific cysteine[27]. The cryptophyte *G. theta* has been shown to possess members of each family of bilin lyase including an S-type lyase which is specific for attaching a chromophore to Cys82[28]. The S-type lyases transfer chromophores to a folded apo-PBP[29]. Our data indicate that the substrate for the Cys82 lyase is the folded apo αβ protomer, as the N-terminus of the α subunit confers chromophore specificity. It remains to be discovered whether *H. andersenii* possesses two S-type lyases, one for PCB and one for PEB, given the asymmetry in *Ha*PE645 (Supplementary Note 6).

**Table 1 X-ray crystallographic data reduction and refinement.**

| Data collection<br>All (Outer Shell) | 8EL3<br>*Ha*PE555 | 8EL4<br>*Ha*PE555 | 8EL5<br>*Ha*PE555 | 8EL6<br>*Ha*PE555 | 7SSF<br>*Ha*PE560 | 7SUT<br>*Ha*PE645 |
|---|---|---|---|---|---|---|
| Space group | P 1 21 1 | P 1 21 1 | P 1 21 1 | P 1 21 1 | I 1 2 1 | P 1 21 1 |
| Cell dimensions | | | | | | |
| a, b, c (Å) | 64.80, 76.84, 103.36 | 63.53, 71.26, 48.28 | 64.86, 75.60, 99.36 | 61.71, 70.00, 48.02 | 84.32, 67.97, 184.50 | 54.03, 80.98, 115.84 |
| α, β, γ (°) | 90, 110.73, 90 | 90, 108.47, 90 | 90, 110.30, 90 | 90, 110.31, 90 | 90, 99.33, 90 | 90, 92.21, 90 |
| Resolution (Å) | 1.57 (1.60 -1.57) | 1.73 (1.76-1.73) | 1.67 (1.70-1.67) | 1.83 (1.87-1.83) | 1.45 (1.47-1.45) | 1.49 (1.52-1.49) |
| $R_{sym}$ or $R_{merge}$ | 0.095 (1.399) | 0.135 (2.797) | 0.091 (2.595) | 0.21 (1.539) | 0.095 (1.424) | 0.056 (1.117) |
| I / σI | 10.5 (1.0) | 8.2 (0.7) | 9.5 (0.7) | 6.9 (1.2) | 12.5 (1.4) | 9.9 (0.9) |
| Completeness (%) | 98.2 (81.9) | 100 (99.9) | 100 (99.7) | 100 (100) | 99.8 (99.3) | 93.4 (58.3) |
| Redundancy | 6.4 (4.3) | 6.5 (6.7) | 6.7 (6.1) | 7.3 (7.3) | 7.4 (7.2) | 3.6 (2.2) |
| **Refinement** | | | | | | |
| Resolution (Å) | 1.57 | 1.73 | 1.67 | 1.95 | 1.45 | 1.49 |
| No. unique reflections | 129,756 (5326) | 42,700 (2325) | 104,433 (5125) | 33,901 (2072) | 181,639 (8893) | 151,716 (4645) |
| $R_{work}$ / $R_{free}$ | 0.1701 / 0.2010 | 0.1877 / 0.2392 | 0.1858 / 0.2314 | 0.1814 / 0.2554 | 0.1541 / 0.1921 | 0.1455 / 0.1747 |
| No. atoms | | | | | | |
| Protein | 17,949 | 9386 | 19,550 | 8437 | 15,495 | 14,867 |
| Ligand/ion | 1596 | 798 | 1676 | 798 | 1571 | 1443 |
| Water | 695 | 192 | 340 | 256 | 816 | 456 |
| B-factors (Å$^{-2}$) | 23.34 | 35.85 | 39.11 | 22.16 | 23.87 | 26.94 |
| Protein | 23.21 | 36.19 | 39.55 | 22.41 | 23.81 | 26.80 |
| Ligand/ion | 20.60 | 32.60 | 35.27 | 20.06 | 19.44 | 25.99 |
| Water | 28.49 | 35.48 | 37.56 | 21.57 | 28.93 | 30.69 |
| R.M.S. deviations | | | | | | |
| Bond lengths (Å) | 0.013 | 0.013 | 0.013 | 0.013 | 0.013 | 0.011 |
| Bond angles (°) | 1.385 | 1.415 | 1.446 | 1.504 | 1.415 | 1.317 |

A recent study of photoacclimation in two cryptophyte species has shown that the absorbance spectra of the PBPs are altered when cells are photoacclimated via growth under restrictive light conditions using spectral filters[26]. This study concluded that the spectral changes must result from differential chromophorylation of the PBPs which is consistent with the existence of a pool of lyases capable of modifying the attached chromophores as observed in our structure of *Ha*PE645.

**Organization of the soluble cryptophyte light harvesting antenna in the thylakoid lumen.** Given the presence of at least three distinct PBP components in *H. andersenii*, what is the organization of this antenna? Electron micrographs (using high pressure freezing and freeze substitution) of the plastids of *H. andersenii* show clear striations of electron opaque material between the thylakoid membranes indicating densely-packed protein, as observed for other cryptophytes[30–32] (Fig. 4a). The width of the electron opaque material within the thylakoid lumen is estimated to be $12.7 \pm 2.9$ nm (Fig. 4b), which is within the 10–50 nm range of previous measurements (where the width increases with the inverse of light intensity during culturing)[30–32]. Based on the dimensions of the proteins from our crystal structures, we estimate that only around 3–4 proteins (or up to 10–13 proteins in very low light[30–32]) would fit across the thylakoid lumen (Fig. 4c; Methods).

Fluorescence recovery after photobleaching (FRAP) experiments attest to the density and restricted mobility of protein in the thylakoid lumen[33,34]. After photobleaching, the PBP fluorescence either did not recover (no diffusion of PBP over several minutes; *Rhodomonas salina*)[34] or recovered very slowly giving a diffusion coefficient of $(1.5 \pm 0.1) \times 10^{-9}$ cm$^2$/s which is substantially smaller than that of comparable proteins in other cellular compartments (*Rhodomonas* CS24)[33].

All the results above paint a picture of a tightly packed, energetically complex, multi-subunit protein antenna embedded in the thylakoid lumen of the cryptophyte *H. andersenii* where the predominant PBPs are *Ha*PE555, *Ha*PE560 and *Ha*PE645 in the ratio of 5:1:1. From this, how does the antenna organize itself within the thylakoid lumen for robust and efficient energy transport? We have shown that *Ha*PE555 and *Ha*PE560 form the bulk of the light harvesting antenna (roughly 6/7 of the PBPs) and capture the highest energy photons. Absorption and fluorescence derived from excitation-emission maps suggest that *Ha*PE555/560 spectrotypes can transfer excitations between each other as they have substantial spectral overlap (Figs. 1f and 4f Step 2). The absorption of *Ha*PE555 and *Ha*PE560 lie in a spectral region transparent to the chlorophylls that make up the integral membrane photosystems (Fig. 4d)[35]. Thus, *Ha*PE555 and *Ha*PE560 have evolved to capture photons that would be otherwise lost by the integral membrane photosystems: they plug a spectral hole.

However, the integral membrane photosystems of cryptophytes have very little spectral overlap with the fluorescence of *Ha*PE555/560 (Fig. 4e), thus, *Ha*PE555 and *Ha*PE560 are unlikely to transfer energy directly to the membrane systems. Hence, another element is required to bridge the energy gap between the bulk antenna (*Ha*PE555 and *Ha*PE560) and the integral membrane photosystems.

The spectral properties of *Ha*PE645, a much smaller constituent fraction of the cryptophyte PBP antenna (~1/7 of the PBPs), fulfil the requirements for a terminal acceptor of the soluble PBP antenna that acts as the adaptor to bridge the energy gap and transfer the excitations to the integral membrane photosystems embedded in the thylakoid membrane which have a local absorption maximum at $668 \pm 3$ nm (Fig. 4d, e green spectrum).

Having an adaptor protein (*Ha*PE645) coupling the soluble and membrane bound antennas presents an organizational challenge. Once the excitation is transferred to *Ha*PE645 it must then be transferred to a photosystem (or LHC) on the order of a

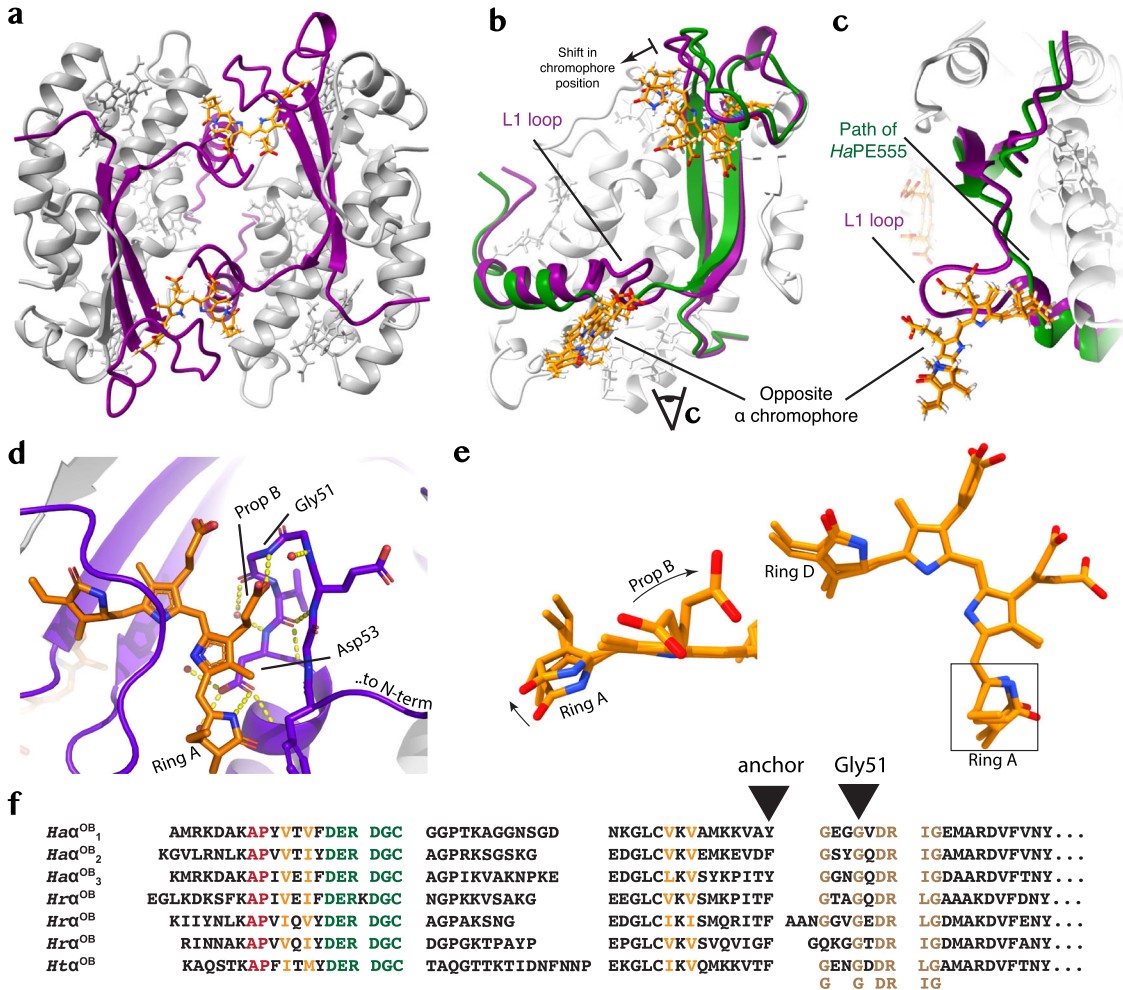

**Fig. 2 The crystal structure *Ha*PE560. a** Cartoon diagram showing the crystal structure of the *open-braced* quaternary form *Ha*PE560 (α subunits purple, β subunits gray). **b** Comparison of the *open-braced Ha*PE560 α subunit structure with that of the *open* form *Ha*PE555. The *open-braced* α subunit is shown in purple cartoon while the *open* form α subunit is in green. The *L1* loop can be seen in the center of the panel. **c**. Inset comparing the *L1* protrusion (marked) and the path of the *Ha*PE555 (green) as viewed by the eye marker in **b**. The chromophores of both proteins are shown in gold in **b** and **c**, with the chromophore of *Ha*PE560 shifted away from the binding pocket. **d** The interaction between the *L1* loop and the neighboring α chromophore. The seven residue *L1* loop interacts with the α19 chromophore of the opposite α subunit in two key places: Asp53, which ligates the nitrogen of pyrrole ring A and Gly51, where the propionate from pyrrole ring B forms a hydrogen bond to the backbone nitrogen, stabilizing the β hairpin structure. **e** The interaction between the *L1* loop and the α19 chromophore from the opposite α subunit leads to: a rotation of pyrrole ring A relative to ring B, when compared to *open* form structures; and an anchoring of the propionate side chain of pyrrole ring B, decreasing flexibility (and thus excitation decay routes). Right panel shows the complete α19 chromophore in the same orientation as **d**, while the left panel highlights the changes in chromophore geometry. **f** An alignment of *open-braced* form α subunit transcriptome sequences from: *Ha – H. andersenii*; *Hr – H. rufescens*; and *Ht – H. tepida*. Color codes are as per Fig. 1a.

nanosecond[36] otherwise it risks energy loss. As the distance of the *Ha*PE645 to the membrane bound antennas increases, the transition probability for excitation transfer between *Ha*PE645 and the membrane system decreases, increasing the likelihood of trapping and fluorescence on *Ha*PE645 (Fig. 4). This implies that for maximal efficiency, *Ha*PE645 should be near the thylakoid membrane which contains both photosystems and integral membrane LHC antennas (Supplementary Fig. 9; Supplementary Notes 7 and 8). Under the expected range of physiological conditions 50% FRET efficiency is achieved when *Ha*PE645 is either adjacent or better, tethered to the thylakoid membrane.

Based on this evidence, we speculate that under normal physiological conditions *Ha*PE645 should be the closest protein to the membrane (Supplementary Fig. 9) so as to ensure maximal energy transfer from the bulk antenna to the membrane systems. It may be significant that *Ha*PE645 maintains a distinct *closed* quaternary structure, either as a source of molecular recognition or due to its functional properties. We note that the recent

structure of a cryptophyte photosystem I supercomplex has an unknown protein protruding into the lumenal space, potentially providing a membrane-proximal binding site for an adaptor such as *Ha*PE645, potentially orienting it for favorable transfer from the β82 PCB chromophore[37]. The bulk antenna proteins, *Ha*PE555 and *Ha*PE560, are densely packed to form a concentrated glass of proteins as evidenced by electron micrographs and FRAP experiments. It is also possible that the rafts of *Ha*PE555 filaments seen in all crystal forms are part of the organizing principle of the antenna architecture (Supplementary Figs 10–12; Supplementary Notes 3 and 8).

Based on this, a picture for light harvesting emerges (Fig. 4f). Sunlight filters through water and visible photons are captured by the bulk antenna proteins *Ha*PE555 and *Ha*PE560 (Fig. 4f, Step 1). Given their spectral properties, energy is transferred readily among these proteins (Fig. 4f, Step 2). Eventually, the excitation is transferred to the *Ha*PE645 adaptor protein (Fig. 4f, Step 3). Within this protein the excitation is transferred to the β82 PCB

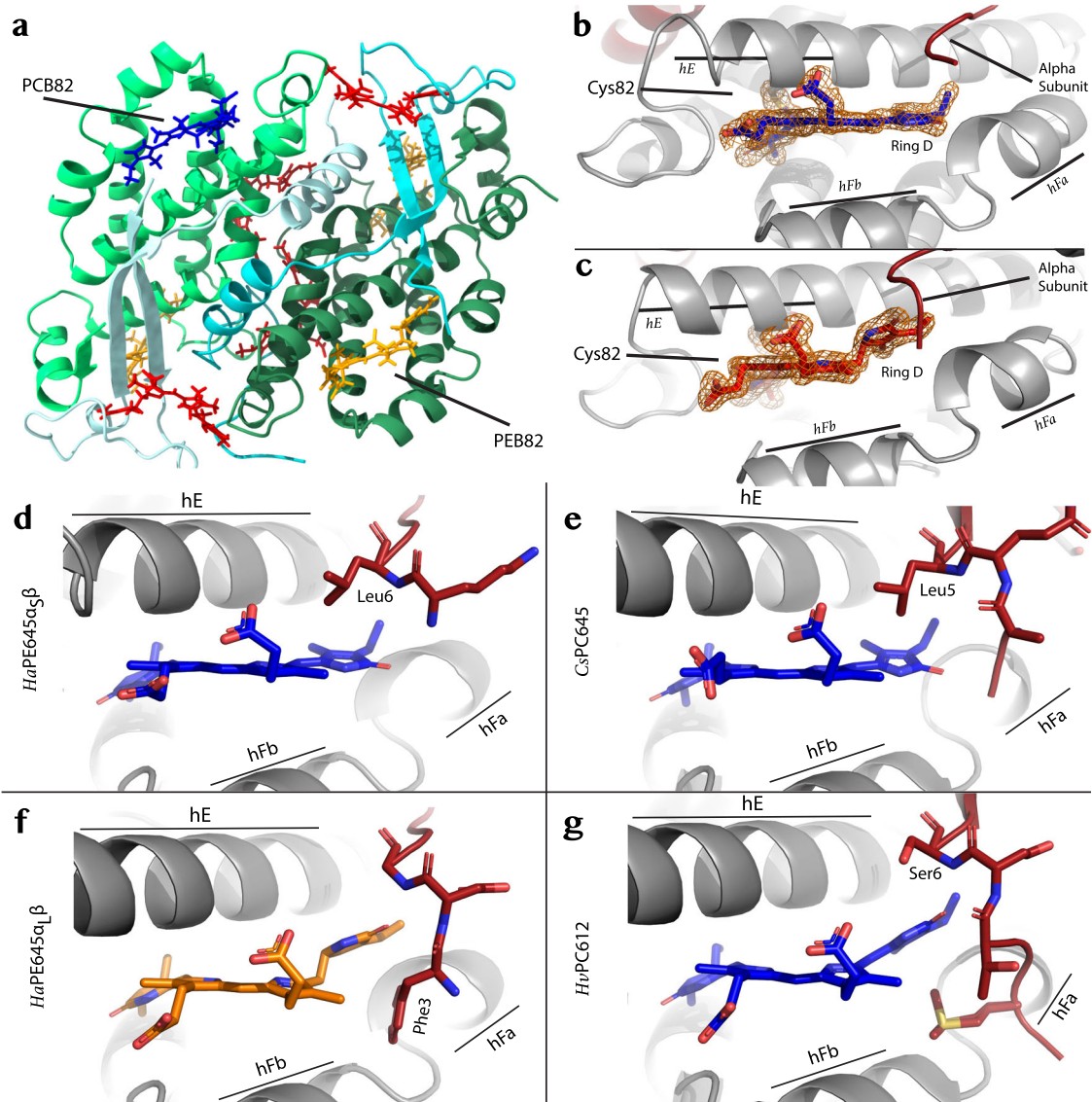

**Fig. 3 The crystal structure of _Ha_PE645. a** Cartoon representation of the _closed_ form _Ha_PE645 structure ($\alpha_L$ cyan; $\alpha_S$ pale cyan; corresponding β subunits forest and lime green, respectively). **b** The electron density (orange mesh) for the β82 chromophore attached to the $\alpha_S$β protomer clearly shows that it is PCB, where pyrrole ring D is part of the conjugated structure. **c** In contrast, the electron density (orange mesh) for the β82 chromophore attached to the $\alpha_L$β protomer clearly shows that it is PEB, where pyrrole ring D is separated by two sp3 carbon atoms from the conjugated ring structure. **d** Shows the interactions between the N-terminus of the $\alpha_S$ subunit and the PCB β82 chromophore in _Ha_PE645, which are similar to those seen in **e** for the PCB β82 chromophore in _Cs_PC645. **f** Shows the interactions between the N-terminus of the $\alpha_L$ subunit and the PEB β82 chromophore in _Ha_PE645, which are similar to those seen in **g** for the PCB β82 chromophore in _Hv_PC612.

chromophore on the $\alpha_S$β protomer (Fig. 4f, Step 3, with panel C showing internal transfer to PCB). From there, the excitation is transferred to the integral membrane photosystems (Fig. 4f, Step 4).

Evidence of a cryptophyte antenna with multiple expressible (and expressed) cryptophyte α chains exists in the literature[17,23] but has been seemingly buried until recently[24] (Supplementary Note 9). In fact, in organisms with a majority ~550 nm absorbing PBP, evidence of a 645 nm absorbing protein is present but often discarded as its significance was not fully grasped[11,26,38,39]. Furthermore, evidence of simultaneous expression of _open_ and _closed_ forms has been recently discovered in _H. virescens_[10].

## Conclusions

In _H. andersenii_, we have discovered an energetically complex, multi-subunit antenna comprised of both _closed_ and _open_ form PBPs, presenting a model for the complete antenna. Given

transcriptomics for other cryptophytes[10] as well as spectroscopy describing proteins with a 645 nm peak, our model is likely more general for cryptophyte light harvesting, with many cryptophytes having a similar rainbow of proteins (Supplementary Note 10). We envisage that a typical cryptophyte soluble antenna would comprise a bulk light harvesting protein with a peppering of other PBPs with specific spectral, structural, and spatial properties so as to energetically couple the soluble antenna to the integral membrane photosystems.

## Methods

**Transcriptomics sequences.** Complete transcriptomes of all four _H. andersenii_ strains[40] (MMETSP0043 (strain CCMP644), MMETSP1041 (strain CCMP439), MMETSP1042 (strain CCMP1180), MMETSP1043 (strain CCMP441)) were download from https://www.imicrobe.us/#/projects/104. The transcriptomes

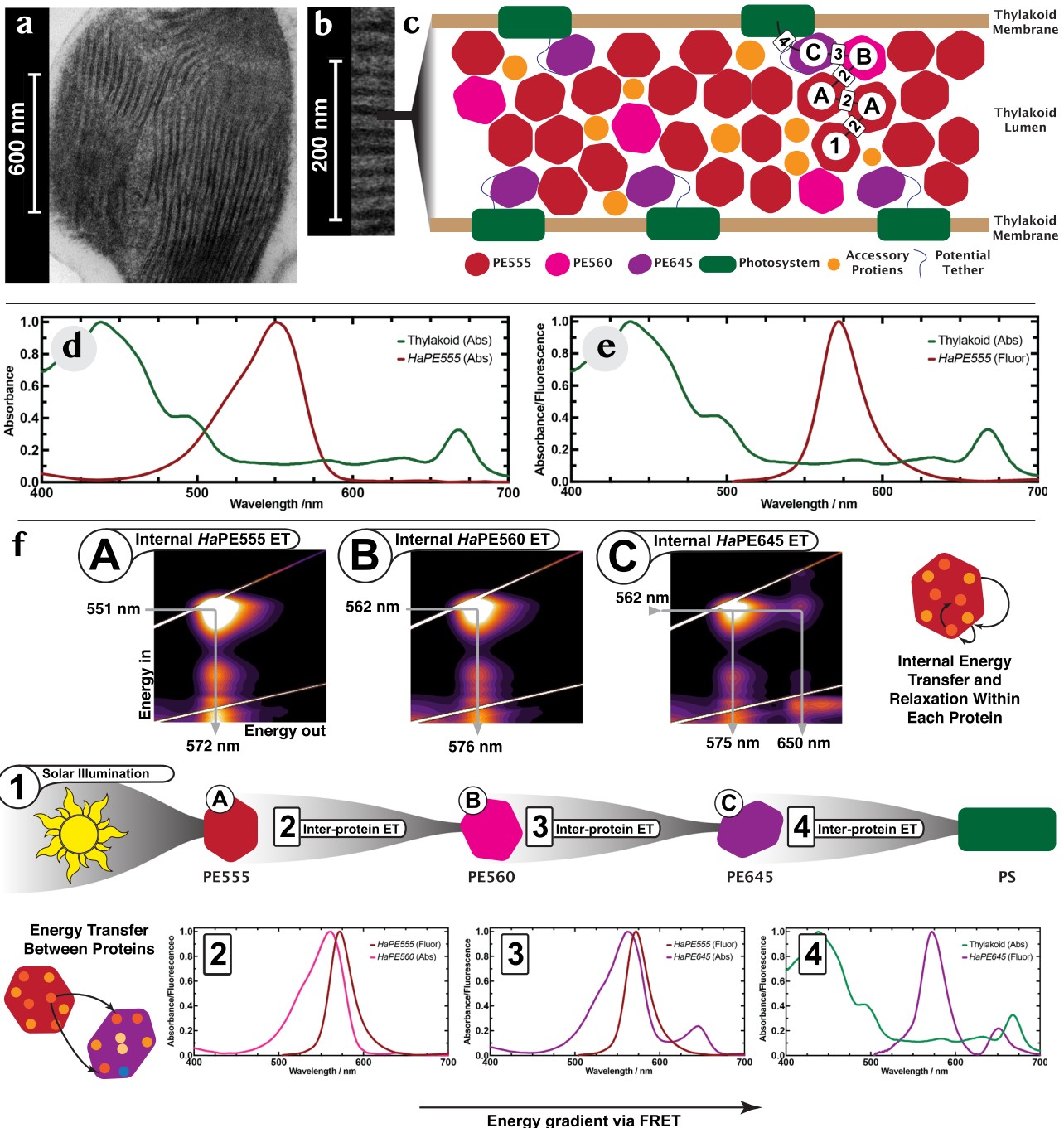

were packaged into databases using Genome workbench within which a custom BLAST search was done against published α chain sequences from cryptophyte PBP structures on the PDB (Accession codes: 4LMX, 4LMS, 4LM6, 1XG0). The resulting list of 154 sequences produced was then manually pruned using the known cryptophyte α sequence motif AP-x(9–10)-C to 67 sequences. The mature proteins were then generated using the cut-site motif: AXA[18]. There does not appear to be any correlation between the cut site (AxA) and any other mature sequence features. To remove redundancy within the transcriptomes from different strains, we chose the most complete read (longest and least ambiguous) for each sequence across the transcriptomes manually. The resulting list of 22 sequences (Supplementary Table 1) was aligned using MAFFT[41] to a structural alignment of all published cryptophyte α subunits (Fig. 1a).

During this process, one sequence stood out and did not align correctly with the other sequences. This outlier was a tandem sequence comprised of two concatenated α sequences. This transcript read however does have an ambiguous region (denoted by X in the sequence) and in-between the two motifs there exists a predicted ADA cut site which would produce a mature α subunit from the C-terminal copy. In some transcriptomes, this tandem sequence appears to have been split into two separate reads and as such the tandem sequence has been treated as non-biological in this case.

Three peptides ($Ha\alpha^O_{10}$, $Ha\alpha^C_7$, $Ha\alpha^C_8$) had longer than usual mature N-terminal tails. Furthermore, $Ha\alpha^C_8$ did not have a discernible AxA cut site, but rather a SFS cut site (as demarcated by TargetP[42]). The *closed* form peptides $Ha\alpha^C_7$ and $Ha\alpha^C_8$, as well as $Ha\alpha^O_{11}$ had longer than usual mature C-terminal tails. In

**Fig. 4 The organization of the soluble cryptophyte light harvesting antenna in the thylakoid lumen. a** Electron micrograph of a thin section from a whole freeze-substituted *H. andersenii* cell (overview). **b** Close up of the chloroplast showing clear striations of dense material in the thylakoid lumen. **c** A model of the soluble light harvesting antenna residing in the lumenal space of the thylakoid. *Ha*PE645 (purple) resides near the membrane while the lumenal space is filled with a dense phase of *Ha*PE555 and *Ha*PE560 (red and pink, respectively). A typical energy transfer path is shown as a series of black lines connecting PBPs to the integral membrane photosystems (green). The capital letters and numbers along this path refer to energy transfer processes as described in **f. d** The absorption spectrum of the major soluble cryptophyte antenna protein *Ha*PE555 (red) plugs a spectra hole in the absorption spectrum of the photosystems and antennas in the thylakoid membrane (green). **e** The fluorescence emission of *Ha*PE555 (red) shows minimal spectral overlap with the absorption spectrum of the thylakoid membrane system (green) where chlorophyll is the main chromophore in the integral membrane antennas and photosystems. **f** Proposed light harvesting pathway shown as a series of excitation-emission maps (**A-C**) demonstrating internal excitation transfer within PBPs with associated 1D spectra showing overlap between emission and excitation (**2–4**) for inter molecular energy transfer between proteins. This pathway corresponds to that shown in **c**. **Step 1**: sunlight is incident on *Ha*PE555 which absorbs around 551 nm. Energy transfer moves the excitation within the protein (excitation-emission map **A**). **Step 2**: energy is transferred to either *Ha*PE555 or *Ha*PE560 as the emission spectra of *Ha*PE555/*Ha*PE560 have substantial spectral overlap with the absorption spectra of *Ha*PE555/*Ha*PE560 (spectrum **2** shows overlap of emission of *Ha*PE555 (red) and absorption of *Ha*PE560 (pink)). The excitation moves within *Ha*PE560 (excitation-emission map **B**). **Step 3**: energy is transferred from either *Ha*PE555 or *Ha*PE560 to *Ha*PE645 as their emission spectra **3** (red, only *Ha*PE555 shown) have substantial spectral overlap with the absorption spectrum of *Ha*PE645 (purple). Within *Ha*PE645, the excitation migrates to the final acceptor PCB β82 chromophore (excitation-emission map **C**). **Step 4**: spectral overlap between *Ha*PE645 emission (purple) and photosystem absorption (green) facilitates energy transfer to the integral membrane system. The location of the *Ha*PE645 adaptor that couples the soluble antenna to the integral membrane systems must be close to the membrane for efficient transport (**c**).

---

the alignment (Fig. 1a), these N- and C-terminal extensions were truncated but noted with an ellipsis (with the number of residues removed being indicated).

For the analysis of the distribution of the L1 loop in *open* form proteins within the *Hemiselmis* genus, the transcriptomes MMETSP1355 (*Hemiselmis tepida*, Strain CCMP443), MMETSP1357 (*Hemiselmis rufescens*, Strain PCC563) and MMETSP1356 – (*Hemiselmis virescens*, Strain PCC157) were downloaded and the same analysis was performed as above.

***Hemiselmis andersenii* cultures.** Cultures of *H. andersenii* strain CCMP644 were obtained from the National Center for Marine Algae and Microbiota (Bigelow Laboratory for Ocean Sciences) and grown continuously in a temperature range between 23 °C and 19 °C illuminated by Osram daylight fluorescent tubes with an average radiant power of 1.2 Wm$^{-2}$ at the cultures. Cultures were grown in rehydrated sea salt (Red Sea brand; 33.5 g/L in MilliQ water) with 0.5 mL/L of 2000x concentrated silicate free f/2 trace elements from Algaboost and 0.25 mL/L of 1,000,000× f/2 vitamin mix from Algaboost. Cells were passaged every fortnight into fresh media in a roughly 1:200 ratio of cell inoculant to media. Fresh media was not autoclaved prior to use. Cells were grown in 500 mL batches in 2 L conical flasks with microporous cloth taped to the opening and agitated by lightly swirling every week. Cells were harvested at the same time point as inoculation of new cultures. This was achieved by centrifugation at 2000 g for 40 mins at 4 °C (400 mL centrifuge bottles in a F10BCI-6x500y rotor and Avanti J-E centrifuge from Beckman Coulter). Samples were frozen and stored at −80 °C.

**Protein purification.** Cell pellets were resuspended using a magnetic stirrer at 4 °C in 50 mL of 25 mM HEPES and 100 mM NaCl buffer at pH 7.5 with a ~2 mg of lysozyme, ~50 µg DNAse 1 and one pellet cOmplete EDTA-free Protease Inhibitor Cocktail for 30 mins. The resulting slurry was passed through the Constant Systems cell disruptor three times at a setting of 30 kPa. The homogenized lysate was then sedimented at 48,000 g for 1 hr and 4 °C in an Avanti J-E centrifuge (Beckman Coutler) in the 50 mL tubes with a rotor JA25.50 (Beckman Coulter). The supernatant was mixed in a 1:1 ratio of 100% saturated ammonium sulphate (at 4 °C) and stirred for 1 hr at 4 °C. This mixture was sedimented by centrifugation at 40,000 g and 4 °C for 1 h (400 mL centrifuge bottles in a F10BCI-6x500y rotor and Avanti J-E centrifuge from Beckman Coulter). The supernatant was concentrated using 30

kD Amicon centrifuge concentrators. The sample buffer was exchanged by diluting the concentrated supernatant in a 1:10 ratio with 25 mM HEPES buffer at pH 7.5, then reconcentrating it and repeating this procedure five times.

The concentrated sample was then fractionated using anion exchange chromatography in a HiTrap Q HP 5 mL (GE Life Science) column equilibrated in 25 mM HEPES at pH 7.5 eluted using a 0–0.35 M NaCl gradient at 1 mL/min (Supplementary Fig. 1, lower chromatogram). Two fractions were separated from this step; a pink fraction with everything that eluted before 110 mL retention volume (including the unbound fraction) and a purple fraction including everything after.

The pink and purple fractions were both independently passed through a cation exchange step in a HiTrap SP HP 5 mL (GE Life Science) equilibrated in 20 mM sodium acetate at pH 5.0 and eluted using a 0–0.25 M NaCl gradient at 1 mL/min (Fig. 1b and Supplementary Fig. 1, upper chromatograms). Fractions were extracted as 3–4 mL samples around each absorption peak in each chromatogram.

The largest samples (PE555A, PE555B, PE560A, PE645A) were then purified using size exclusion chromatography using a Superdex GL 200 Increase 10/300 (GE Life Science) column run at 4 °C in 20 mM HEPES, 100 mM NaCl and 1 mM NaN3 at pH 7.5 run over 1.5 column volumes after the column was equilibrated using 1.5 column volumes of the running solution. Resulting samples were finally concentrated using 30 kDa Amicon centrifuge concentrators. Samples were frozen and stored in a −80 °C freezer. Samples used for experiments were kept in this buffer unless otherwise stated.

**Thylakoid membrane protein purification.** Cell pellets were resuspended using a magnetic stirrer at 4 °C in 50 mL of 25 mM HEPES and 100 mM NaCl buffer at pH 7.5 with a ~2 mg of lysozyme, ~50 µg DNase I and one pellet cOmplete EDTA-free Protease Inhibitor Cocktail for 35 mins. The resulting slurry was passed through the Constant Systems cell disruptor three times at a setting of 30 kPa. This was centrifuged at 5000 x g for 10 mins at 4 °C (400 mL centrifuge bottles in a F10BCI-6x500y rotor and Avanti J-E centrifuge from Beckman Coulter). The supernatant was then centrifuged at 12,000 g for 10 mins at 4 °C using the JA.25.50 fixed-angel rotor in the same centrifuge. The pellet containing thylakoid membranes was resuspended in 0.33 M sorbitol, 25 mM Hepes, 20 mM KCl, 5 mM EDTA, at pH 7.8. This was then centrifuged for 30 min at 38,000 g using the same rotor as the previous step. The residual phycobiliproteins were removed

by centrifugation of the resuspended pellet two times in the aforementioned buffer and discarding the supernatant. Finally the pellet was dissolved in 0.1% v/v Triton X-100, 0.33 M sorbitol, 25 mM Hepes, 20 mM KCl, 5 mM EDTA, at pH 7.8 so as to solubilize the thylakoid integral membrane proteins. The spectrum was collected using SPECTROstar Nano instrument in a 1 cm quartz cuvette.

**Linear absorption spectroscopy.** Absorption spectra were taken in quadruplicate for each protein analyzed using the Nanodrop by Thermo Fisher. Spectra were then averaged and normalized to the area under the curve using Mathematica.

**LC-MS/MS.** An appropriate amount of sample was reduced with 5 mM dithiothreitol, 10 mM iodoacetamide followed by tryptic digestion overnight at 37 °C. Digested peptides were desalted with C18 stage tip (Thermo Fisher) before being separated by nano-LC using an Ultimate 3000 HPLC and autosampler system (Dionex, Amsterdam, Netherlands). Samples (2.5 μl) were concentrated and desalted onto a micro C18 precolumn (300 μm × 5 mm, Dionex) with $H_2O$:$CH_3CN$ (98:2, 0.05 % TFA) at 15 μl/min. After a 4 min wash the pre column was switched (Valco 10 port valve, Dionex) into line with a fritless nano column (75 μ × ~ 10 cm) containing C18 media (1.9 μ, 120 Å, Dr Maisch, Ammerbuch-Entringen Germany). Peptides were eluted using a linear gradient of $H_2O$:$CH_3CN$ (98:2, 0.1 % formic acid) to $H_2O$:$CH_3CN$ (64:36, 0.1 % formic acid) at 200 nL/min over 30 min. High voltage (2000 V) was applied to low volume tee (Upchurch Scientific) and the column tip positioned ~0.5 cm from the heated capillary (T = 275 °C) of an Orbitrap Velos (Thermo Electron, Bremen, Germany) mass spectrometer. Positive ions were generated by electrospray and the Orbitrap operated in data dependent acquisition mode. A survey scan m/z 350–1750 was acquired in the Orbitrap (Resolution = 30,000 at m/z 400, with an accumulation target value of 1,000,000 ions) with lockmass enabled. Up to the 10 most abundant ions (>4000 counts) with charge states > +2 were sequentially isolated and fragmented within the linear ion trap using collisionally induced dissociation with an activation q = 0.25 and activation time of 30 ms at a target value of 30,000 ions. M/z ratios selected for MS/ MS were dynamically excluded for 30 s. All MS/MS spectra were searched against UniProt and customized database with MASCOT (version 2.3)[43] with the following search criteria: enzyme specificity was trypsin; precursor and product ion tolerances were at 4 ppm and ± 0.4 Da, respectively; variable modification of methionine oxidation; and one missed cleavage was allowed. Mass spectrometric analysis was carried out at the Bioanalytical Mass Spectrometry Facility, University of New South Wales, Australia.

**Electrospray ionization spectrometry.** Samples were dialyzed into water to remove any traces of salt in the sample. Samples were fractionated through reversed-phase chromatography on a ACQUITY UPLC Protein BEH C4, 300 Å, 1.7 μm, 2.1 mm × 50 mm column (Waters) with a starting buffer of 0.1% formic acid in water (v/v) and final buffer of 0.1% formic acid in acetonitrile (v/v). Electrospray ionization was performed in-line by a Vion QTof mass spectrometer (Waters) using ESI positive sensitivity mode in a mass range of m/z 500–4000 Da. Data were deconvolved using UNIFI (Waters). All equipment is part of the Bioanalytical Mass Spectrometry Facility, Mark Wainwright Analytical Centre, UNSW.

Theoretical masses for all mature α sequences derived from the transcriptome were calculated using *SwissProt* and the mass of a phycoerythrobilin (PEB) chromophore (586.7 Da) was added by hand. Note that some sequences could not be given a mass estimate as they contained either: misreads (X in the sequence); they did not contain a start site; or some unknown mature peptides may have a different chromophore bound that is not PEB. When interpreting masses, ladders of 16 Da are observed above the native mass on peptides with multiple methionines, suggesting methionine oxidization. There are mass ambiguities here also given the similarity in size of many α subunits to the resolution of the mass spectrometry setup.

From both the peptide identification by fragment LC-MS/MS and the intact electrospray mass spectrometry, this is by no means an exhaustive analysis. The isolates analyzed were chosen as they were the only ones with a high enough concentration and purity. From chromatograms during purification, it was clear there were further isolates available, but at a concentration and purity that was too low. The mass spectrometry that has been done therefore represents a survey of the most prominent α subunit pairings and was somewhat hampered by cross-contamination between different isolates taken from cation exchange.

**Protein abundance calculations.** Spectrotype abundance estimates were calculated by first estimating the fraction of pink vs purple proteins and then estimating the subdivisions within each spectrotype. The fraction of pink vs purple was calculated via integration under the curve of the 560 nm absorbance trace in Mathematica and cut at the same site as per purification instructions above. Due to a non-linearity in the 560 nm absorbance arising from saturation of the detector, the chromatogram trace measured at 280 nm absorbance (which was not saturated) was used as a substitute by manually scaling the 280 nm peak (unbound fraction) to the 560 nm absorbance around this peak where the 560 nm signal was small enough to be in the linear regime of the detector. This matching was done by eye. An uncertainty in the scaling was estimated from this procedure based on the limits of a satisfactory fit. Integration of the 560 nm absorbance of the purple fraction was scaled by 7/8 to accommodate the chromophore alteration. The areas under the curve of the pink and purple fractions were compared as a percentage of the total area. Within the pink or purple fractions, the most prominent peaks of the 560 nm absorption in the cation exchange chromatogram for each species (HaPE555A and HaPE560A for pink; HaPE645A for purple) were fit to a Lorentzian-like function (below; with ε a free parameter) using NonlinearModelFit in Mathematica. The area under the curve was then compared to the total area under the curve of the 560 nm absorption to produce a percentage fraction (Supplementary Table 2). The parameters in the equation below are: A — height scaling, Γ— peak width, a — elution peak position, x — elution volume and ε — non-Lorentzian factor.

$$\frac{A}{\pi} \frac{\frac{1}{2}\Gamma}{|x - a|^{2+\varepsilon} + \left(\frac{1}{2}\Gamma\right)^2} \tag{1}$$

**Excitation-emission maps.** Spectra were collected using a Jasco FP-8500 Fluorescence Spectrometer and plots were produced in Mathematica. Truncation of the data at a particular height (for heat maps) was also performed in Mathematica.

**Measuring concentrations of PBPs for crystallography.** Due to the chromophores, it is not possible to sensibly use UV absorption at 280 nm to estimate protein concentrations for PBPs. As the most stable feature is the optical absorbance from the chromophores, we instead estimate concentration by measuring the absorbance at 560 nm. This provides a useful guide for crystallization experiments, however, the unit is not calibrated in terms of molar concentration.

**Crystal structures of *Ha*PE555.** *Ha*PE555 fractions were crystallized using a sitting drop and vapor diffusion method in a 96 well plates at room temperature. The crystal for 8EL6 was grown in PEG3350 23.6% (w/v). The crystals for 8EL3–8EL5 were grown in PEG3350 25% (w/v) + AddScreen by Hampton Research. Specifically, PEG3350 25% (w/v) + NaBr 0.01 M (8EL3), PEG3350 25% (w/v) + Sarcosine 0.01 M (8EL4) and PEG3350 25% (w/v) + Benzamidine HCl 2% (w/v) (8EL5). All drops were 150nL protein and 150nL mother liquor with protein concentration of 0.7 absorbance units at 560 nm with 0.1 mm path length and set by Formulatrix NT-8 and Art Robins Phoenix crystallization robots.

X-ray diffraction datasets were collected at Australian Synchrotron beamlines MX1 and MX2. The experiment number (as part of the Sydney Collaborative Access Program) for 8EL3-8EL5 is 16068g and the experiment number for 8EL6 is 14590a. Data were collected at a wavelength of 0.9537 Å (13.0 keV) and temperature 100.0 K. Crystals were cryoprotected in 20% glycerol (8EL3-8EL5) or Paratone-N (8EL6).

Data processing for all datasets was performed in DIALS (part of the CCP4 suite; version 7.0.066[44]). Processed data was phased by molecular replacement in phenix.phaser (part of the Phenix suite; version 1.15.2[45]) using the published *H. andersenii* PE555 (PDB 4LMX[9]) with all asymmetry removed. All data was reindexed such that the unit cell vector c was pointing along the filament direction. All models were iteratively built using phenix.refine (part of the Phenix suite; versions 1.15.2-1.19.2[45]) and Coot (versions 0.8.9.2-0.9.5[46]).

In the later part of refinement, a non-standard approach had to be taken to resolve the problem of mixed α chains (microheterogeneity; see Supplementary Fig. 5b). A copy of each α chain (*Ha*α$^O$1 and *Ha*α$^O$2) was modelled into each α subunit position as complete alternate conformer chains with different chain names. For residues that had the same identity, the duplicate was removed. The backbone atoms of each pair of residues in each α subunit were restrained to each other in Phenix using a harmonic restraint with a 0 mean and 0.05 σ. During this procedure, all occupancy refinement was turned off and the occupancy of each alternate chain was estimated heuristically using the B-factor of variable residues. Occupancies were manually varied and the most plausible occupancy split was chosen as 50:50 occupancy as this brought the B-factors of each altimer as close as possible. Following phasing and initial model building, we believed that 8EL6 was simply a symmetric dimer of two *Ha*α$^O$2. Modelling microheterogeneity in the same fashion as 8EL3-8EL5 dispelled the idea that 8EL6 could be symmetric as a 50:50 split seemed to model the density well, not producing any difference density and having equal B-factors. Furthermore, the differences in the C-termini of each alpha chain meant that some waters in the structure had to have occupancies correlated to the alternate chains.

The two α subunits are nearly identical, and the few non-identical residues between the different α subunits do not make any substantial crystallographic lattice contacts. As such, the crystal packing in principle cannot differentiate between different arrangements with α chains swapped (i.e. (α$_1$β).(α$_2$β) versus (α$_2$β).(α$_1$β)) leading to a statistical mixture of two overlayed chains which could be seen in the electron density. The same problem was encountered in the original PE555 structure (PDB 4LMX[9]; Supplementary Figs 2a and 3a). How can an (α$_1$β)(α$_2$β) pairing take place when there are no non-identical residue pairs on the binding interface? The differences in the C-terminal tails of each α subunit, which do make contact, may provide a mechanism for heterodimerization. It is unclear from the crystallography as to whether *Ha*PE555 is a statistical mixture of (α$_1$β).(α$_2$β), (α$_2$β).(α$_1$β), (α$_1$β).(α$_1$β) and (α$_2$β).(α$_2$β) given the near identity of the α subunits (55/67 residues). However, as noted previously[9], it is more likely that each PBP in the crystal is either (α$_1$β).(α$_2$β) or (α$_2$β).(α$_1$β), as the longer α$_1$ would result in a steric clash within a putative (α$_1$β).(α$_1$β) PBP. We note that a similar microheterogeneity has been observed in the recent crystal structure of *Ps*PE545 from *Proteomonas sulcata* where the crystal contains an overlay of (α$_L$β).(α$_S$β) and (α$_S$β).(α$_L$β)[26].

In modelling electron density maps, a lot of negative Fo-Fc regions were spotted.

This corresponded to larger than van der Waals voids that were present in these structures, as well as *Ha*PE560 and *Ha*PE645. It was thought that these were the result of solvent scaling problems (i.e. Phenix has assumed there is bulk solvent given the size of the voids however, no electron density is observed for that region). Ultimately, this was a minor issue in refinement and ignored. There was also a lot of disorganized solvent density that was likely due to partial PEG occupation.

Each α chain contains a 5-hydroxylysine which can be identified clearly in the density and has been noted previous structures of cryptophyte PBPs[9]. The identity of each chromophore was also called as either phycoerythrobilin (PEB) or dihydrobiliverdin; (DBV) based on either sp2 or sp3 bonding within pyrrole ring A, which provides clear difference density if a planar ring is swapped for a non-conjugated ring (or vice versa).

Data reduction and refinements statistics for each structure are shown in Table 1.

**Crystal structure of *Ha*PE560.** *Ha*PE560 was crystallized using a sitting drop and vapor diffusion method in 96 well plates in PEGMME5000 21.4% (w/v) + Tris (pH 6.5) 0.162 M (150 nL protein + 150 nL mother liquor with a protein concentration of 0.7 absorbance units at 560 nm with 0.1 mm path length) at 25 °C where drop-setting was performed by the Art Robins Phoenix robot. Crystals were cryoprotected in 10% glycerol. Data collection for the final structure was performed on the Australia Synchrotron Beamline MX1 (experiment number 14590a; part of the Sydney Collaborative Access Program). Data were collected at a wavelength of 0.9537 Å (13.0 keV) and temperature 100.0 K. Image processing was performed in iMosFLM (part of the CCP4 suite; version 7.0.066[44]) to a resolution of 1.45 Å. Processed data was phased by molecular replacement in phenix.phaser (part of the Phenix suite; version 1.15.2[45] using the already published *H. andersenii* β subunit (chain B from PDB 4LMX, i.e. PE555[9]). The final model was iteratively built using phenix.refine (part of the Phenix suite; versions 1.15.2- 1.19.2[45]) and Coot (versions 0.8.9.2-0.9.5[46]).

Data reduction and refinements statistics are provided in Table 1.

Cryptophyte phycobiliproteins usually support post-translational modifications (other than bound chromophores). As with *Ha*PE555, each α chain contains a 5-hydroxylysine which can be identified clearly in the density and has been noted previous structures of cryptophyte PBPs[9]. A lot of negative Fo-Fc map regions were spotted throughout modelling and were attributed to the same causes as *Ha*PE555.

One curiosity is that, compared to *Ha*PE555, the entire structure of *Ha*PE560 is dilated along one axis and squeezed along a perpendicular axis, both across the solvent filled hole (*Poisson effect*; Supplementary Fig. 4). The RMSD between the two β subunits forming the dimer of protomers of *Ha*PE560 and *Ha*PE555 is 0.98 Å across all atoms and 0.25 Å when only a single β subunit from each protein is used. This *Poisson effect* (dilation correlated to an orthogonal contraction) appears to be due to the *L1* loop which nudges the β subunits apart while drawing the α subunits together in a perpendicular direction as it consolidates the open form.

Evidence for the seven-residue loop insertion in the α subunit was clear in the density (Supplementary Fig. 5a). There were some general regions of the final structure that require care. Firstly, the CD loop of the β subunit is not well ordered but traceable. It is likely that the disorder arises from the fact that the CD loop interacts with a crystallographically neighboring CD loop across a crystallographic two-fold axis. The models for all β chains are also missing the first 2–4 residues as the density was too low to interpret.

At this resolution, some residues began to appear with clearly correlated alternate conformers. Commonly, these were waters, where the alternate conformer of a residue has a water sitting in the primary conformer with occupancy correlated to it. More prominently however, Arg129 of chain F has a secondary conformation which overlaps with a Tris molecule and as such has a correlated occupancy to it. These were all modelled. Some map regions were clearly spaces for more Tris molecules and PEG molecules (in difference density), however, when they were modelled and refined no strong 2Fo-Fc density was present and as such were removed. Ordered solvent was modelled using only the best waters (deleted all waters with e/Å$^3$ < 1.5 or B-factor >50 Å$^2$). There were some densities that were evidently a duet of waters with enough conformational freedom to smear them out into an elongated region. Furthermore, there were densities suggestive of polyethylene glycol as have been extensively seen in other structures. These densities however were not modelled as they were rather weak compared to neighboring modelled regions. This structure also has one modelled chloride ion and perhaps contains a few more that were not modelled. The choice to model this density as a chloride was made as the density was rather electron rich for a water molecule (2.876 e/Å$^3$), the distances to neighboring atoms were too great for water hydrogen bonding and the ligands made sense for chloride (bonding to nitrogen atoms).

Lastly, there were some regions where the density was poor. One region is in the β subunit chain B residues 143–146, which is modelled into anisotropic density. The difference density in this region suggests a translated second copy of this chain is present in this region. This displacement was not prominent enough to model and so was left. A further poor region is residues 28–33 of the α subunit which always seems poor. This region has few constraints and sits atop the α chain chromophore and is solvent exposed. Chain A (α subunit) Asn29 appears to have an alternate conformer, however, due to the disorder of this loop was not modelled. Residue Phe30 of each β subunit, also has poor map-model correlation. This residue makes no crystallographic contacts and is on the surface of the protein and thus disordered.

**Crystal structure of *Ha*PE645**. *Ha*PE645A was crystallized using a sitting drop and vapor diffusion method in 96-well plates and in condition PEG3350 25% (w/v) + 0.1 M Bis-Tris (pH 5.5) (150nL protein + 150nL mother liquor drops with a protein concentration of 0.32 absorbance units at 560 nm with a path length of 0.1 mm) at room temperature. With drop-setting by Formulatrix NT-8 crystallization robot. All gradient optimization trays were produced using the Starlet liquid handling robot from Hamilton. Crystals were cryoprotected with Paratone-N.

Data collection for the final structure was performed on the Australia Synchrotron Beamline MX1 (experiment number 16068e; part of the Sydney Collaborative Access Program). Data were collected at a wavelength of 0.9537 Å (13.0 keV) and temperature 100.0 K. Data processing was performed in DIALS (part of the CCP4 suite; version 7.0.066[44]) to a resolution of 1.49 Å. Processed data was phased by molecular replacement in phenix.phaser (part of the Phenix suite; version 1.15.2[45] using the

already published *H. andersenii* β subunit (chain B from PDB 4LMX, i.e. PE555[9]). The final model was iteratively built using phenix.refine (part of the Phenix suite; versions 1.15.2- 1.19.2[45]) and Coot (versions 0.8.9.2-0.9.5[46]).

Data reduction and refinements statistics are provided in Table 1.

A lot of negative Fo-Fc map regions were spotted through-out modelling and were attributed to the same causes as *Ha*PE555 and *Ha*PE560.

As with *Ha*PE560, there are a few alternate conformers with waters in them. These were not modelled (α subunit Chain A Arg45 and Lys64, β subunit Chain B DBV201 and α subunit Chain E Lys64). There were many high B-factor, anisotropic waters evident in the density. These were largely disregarded as they were low confidence and may have been confounded by low occupancy PEG. The same procedure for water picking was used as with *Ha*PE560. As with *Ha*PE560, a single chloride was also modelled and given the same justification. There were many densities indicative of PEG. Most of these were not modelled as they were either broken, branched or weak. These were however clear in the maps and an attempt was made to model them with only a few being left modelled. PEG molecule (Residue 1 of chain I — PG4 surrounding α subunit chain A Lys78) has some problems in its geometry but is physically reasonable and possibly has a water above the NZ nitrogen. Another PG4 molecule surrounding a water molecule that is hydrogen bonded to the side chain of Asn138 of the β subunit is shown in Supplementary Fig. 5c. There were at least 2 other densities suggestive of Bis-TRIS that were not modelled as such. One high-confidence molecule was modelled and kept. Others were modelled but later removed as there was too much conformational freedom present creating more problems. Some alternate conformers (which were in weak density) were not modelled as they generated clashes. This is largely since the alternate conformers were correlated. Many of these were not modelled as they were minor. Examples include β subunit chain F Ser55 with Met134, β subunit chain F Ile9 with Thr95. As with *Ha*PE560, the N-terminus of the β subunit was disordered (residues 1–15) and was not modelled. The Phe30 of each β subunit, unlike some other structures is not disordered and packs with neighboring Phe30 residues. Phe30 has been seen to interact with neighboring Phe30 residues in the crystals of both *open* and *closed* forms. Lastly one β82 chromophore of each PBP was a PCB instead of a PEB. The density clearly showed a sp2 bonding between the two rings C and D (Fig. 3b). PEB was initially fit to test if biasing the phases would leave the identity ambiguous, however it was clear in the difference density that it could not be PEB.

**Identification of chromophores**. At the resolution of our crystal structures, the identity of each linear tetrapyrrole chromophore can be determined by examining two features: (i) the planarity of pyrrole ring A; and (ii) the co-planarity of the bridging structure between pyrrole rings C and D. For PEB chromophores, pyrrole ring A is non-planar, as two carbon atoms (C2A and C3A) show sp3 hybridization. Thus, the electron density of carbon atoms attached to ring carbon atoms C2A and C3A protrudes above and below the plane of the pyrrole ring. In contrast, for the DBV chromophore, ring A is planar, carbon atoms C2A and C3A show sp2 hybridization and the electron density of the carbon atoms attached to C2A and C3A are coplanar with the rest of pyrrole ring A. Note: sp2 carbon atoms have trigonal planar geometry while sp3 carbon atoms have tetrahedral geometry.

In *Ha*PE645, it is paramount to determine the nature of the chromophores attached to cysteine β82 for the protomer $\alpha_L\beta$ compared to that on $\alpha_S\beta$. In each case, pyrrole ring A is non-

planar, hence the chromophores are either PEB or PCB. To differentiate between these two options, we examined the linkage between pyrrole rings C and D. For PEB, pyrrole ring D is linked to ring C via carbon atom C1D, which shows tetrahedral sp3 hybridization. As such, the linkage between pyrrole rings C and D shows a clear kink in the electron density and these rings are not coplanar. In contrast, for PCB, the equivalent atom shows sp2 hybridization and pyrrole rings C and D are coplanar. Based on this, we can unambiguously assign the chromophore attached to cysteine β82 on the $\alpha_L\beta$ protomer to PEB while that attached to cysteine β82 on the $\alpha_S\beta$ protomer is PCB.

To verify our chromophore assignment with respect to cysteine β82 in *Ha*PE645, we have calculated polder OMIT maps[47] as implemented in the program PHENIX[45] (Supplementary Fig. 8). Supplementary Fig. 8a shows the polder OMIT density for the β82 chromophore on the $\alpha_S\beta$ protomer while Supplementary Fig. 8c shows the atomic model for PCB at this site. The model and its fit to the polder OMIT density are shown in Supplementary Fig. 8e, g, respectively. Similarly, Supplementary Fig. 8b shows the polder OMIT density for β82 chromophore on the $\alpha_L\beta$ protomer. The kink linking pyrrole rings C and D (right hand side) is clear. The PEB model at this site is shown in Supplementary Fig. 8d, while the model and its fit to the polder OMIT density is shown in Supplementary Fig. 8f and h, respectively.

**Chromophore dihedral angle analysis**. Although largely conjugated, the linear tetrapyrrole chromophores deviate from coplanarity between adjacent pyrrole rings due to strain induced by protruding methyl groups. The central pair of rings (B and C) are, in most cases, coplanar at the resolution of the crystal structures and were not analyzed in detail. However, the two outer pyrrole rings (A and D) tend to be twisted with respect to the central pair. To analyze this twisting, we have defined a set of dihedral angles as described in detail in[10]. Briefly, the coordinates for each chromophore were passed into Mathematica and planes were fit to each of the four pyrrole rings. The molecular geometry can be described by a pair of dihedral angles ($\theta_{inner}$, $\theta_{outer}$) starting from the central pyrrole.

**Electron microscopy**. 1 mL samples of *H. andersenii* cells for electron microscopy were taken at the peak of the growth cycle (14 days post inoculation) and prepared by sedimentation at 1000 g for 5 mins in a benchtop microfuge and resuspended to a volume of 250 μL in MilliQ water. To achieve near-to-native state ultrastructure, cells were pelleted at 1000 x g, loaded onto 6 mm gold coated copper high-pressure freezing planchettes (Leica Microsystems) and high-pressure frozen using a Leica EM ICE (Leica Microsystems). Samples were stored in liquid nitrogen and transferred to the automated freeze substitution apparatus Leica EM AFS (Leica Microsystems), containing a solution of 1% osmium tetroxide, 0.2% uranyl acetate (w/v) and 10% water in acetone. Samples were kept at −90 °C for 48 h, slowly warmed to −80 °C (5 °C /h), kept at that temperature for 3 h, warmed to and kept in the same way at −60 °C, −40 °C and −20 °C and finally warmed to 0 °C (5 °C /h). Two washing steps with cold acetone were carried out and the cells were infiltrated overnight with increasing concentrations of Procure resin at room temperature. After 2 changes into fresh Procure resin, the samples were polymerized at 60 °C for 48 h. 70 nm sections were cut with a diamond knife and collected onto carbon coated copper slot grids and post stained with 2% uranyl acetate and lead citrate. Grids were imaged using a Jeol 1400 transmission electron microscope (Tokyo, Japan) operating at 100 kV. To determine the separation between thylakoid membranes, distances between consecutive thylakoid membranes were approximated by using an edge detection algorithm. Images were imported into Mathematica, cropped to show sections with defined striations and aligned so the striations were completely vertical. Using the inbuilt function EdgeDetect and then calculating the distances between consecutive edges, a histogram of distances was generated. From this histogram a mean and standard deviation was produced.

**Fluorescence spectra**. Fluorescence spectra were produced by summing each excitation-emission map data set along the excitation axis to get the total fluorescence. The Rayleigh scattering peaks were removed from the integrated fluorescence spectra by fitting and subtracting off a $\propto r^n$ function to the spectrum using NonlinearModelFit in Mathematica.

All spectra were normalized so the maximum intensity was unity. Graphs were also plotted in Mathematica.

**Protein analysis and graphics**. Protein dimensions were estimated using Pymol[48]. Buried surface area was measured in ChimeraX using the function *measureburiedarea*. This included chromophores but no solvent. All images of protein structures were generated either in PyMol[48] or in ChimeraX[49].

**Statistics and reproducibility**. Chromatographic separation of the three spectrotypes has been performed independently and reliably by two investigators across more than 10 iterations. All absorbance spectra were done in quadruplicate and the average used. Excitation-emission maps were generated once. Each protein was crystallized reproducibly by two independent investigators with sitting drop trays reliably filled with diffraction quality crystals.

**Reporting summary**. Further information on research design is available in the Nature Portfolio Reporting Summary linked to this article.

## Data availability

All coordinates and structure factors have been deposited with the Protein Data Bank under the accession codes: 8EL3, 8EL4 and 8EL5—the filament containing structures of *Ha*PE555 from chromatography peak 555 A; 8EL6—the structure of *Ha*PE555 from chromatography peak 555B; 7SSF - *Ha*PE560; and 7SUT - *Ha*PE645. Supplementary Data 1.xlsx contains source data for the chromatograms in Fig. 1b. Supplementary Data 2.xlsx contains all source data for absorption spectra in Figs. 1c and 4d–f where the columns give the absorption at the wavelength designated in the first row. Each row presents one spectrum as designated in the first column. Rows are grouped to reflect the overlayed spectra in each figure panel. Supplementary Data 3.xlsx contains all source data for PBP fluorescence spectra in Fig. 4e and f where the columns give the fluorescence at the wavelength designated in the first row. Each row presents one spectrum for the protein designated in the first column. Supplementary Data 4.xlsx contains source data for excitation-emission maps in Figs. 1f and 4f. The columns are labelled with the excitation wavelength while the rows are labelled with the emission wavelength. The three excitation-emission maps, one for HaPE555, HaPE560 and HaPE645, are stored as separate sheets in this file. All other data are available in the main text, the Supplementary Information, otherwise, can be obtained from the corresponding author on reasonable request.

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

## Acknowledgements

The authors acknowledge use of facilities in the Structural Biology Facility, the Bioanalytical Mass Spectrometry Facility and the Electron Microscope Unit all within the Mark Wainwright Analytical Centre – UNSW. This research was undertaken in part using the MX1[50] and MX2[51] beamlines at the Australian Synchrotron, part of ANSTO, and made use of the Australian Cancer Research Foundation (ACRF) detector. K.A.M. was supported by AOARD (FA2386-17-1-4101). H.W.R. is supported by a scholarship from the Australian Government Research Training Program. This research was supported by grants from: Australian Research Council Discovery Project grant DP180103964 (P.M.G.C.); United States Air Force Office of Scientific Research AOARD grant FA2386-17-1-4101 (P.M.G.C.); Zelman Cowen Academic Initiatives (P.M.G.C.); Australian Research Council Linkage Infrastructure, Equipment and Facilities grant LE190100165 (P.M.G.C.); DARPA (QuBE) (B.R.G.); and Natural Sciences and Engineering Research Council of Canada ID 4688 (B.R.G.).

## Author contributions

Conceptualization: H.W.R., A.J.L., P.M.G.C., methodology: H.W.R., K.A.M., H.I., B.R.G., P.M.G.C., investigation: H.W.R., A.J.L., K.A.M., H.I., J.B., S.C.G., B.R.G., P.M.G.C., funding acquisition: B.R.G., P.M.G.C., resources: K.A.M., J.B., S.C.G., P.T., B.R.G., P.M.G.C., supervision: K.A.M., P.T., P.M.G.C., writing—original draft: H.W.R., B.R.G., P.M.G.C., writing—review and editing: H.W.R., K.A.M., H.I., B.R.G., P.M.G.C.

## Competing interests

The authors declare no competing interests.
