## [Peer Review File · Communications Biology]

Reviewers' comments:

Reviewer #1 (Remarks to the Author):

This is an extremely impressive paper that describes the antenna system in a cryptophyte alga in remarkable detail. Beginning with sequence analysis, proceeding through protein isolation and identification, structure determination and finally the proposal of a detailed organizational model, the authors have produced a classic study that sets a standard for such work. I enthusiastically recommend that it be published after minor revision.

My main complaint centers around the proposal of a putative tether for the PE645 complex with the photosystem. I think it is reasonable to propose such a tether, as the authors show with the FRET calculations. However, there is not a consistent term used to describe it. In various places it is described as an adaptor, scaffold or tether. The authors should choose one word and be consistent, and also be up front about the fact that there is no direct evidence for it, so it is somewhat speculative.

I am not aware of any time-resolved studies on whole cells of this alga. If there are any, they should be cited, and if not, this provides an ideal way to test the organizational model. Such studies are outside the scope of this paper, as it is already very extensive, but perhaps should be mentioned as a way to test the model.

Reviewer #2 (Remarks to the Author):

This is an insightful paper that I recommend for publication in its present form. It has long been a mystery why there are so many "extra" alpha subunit genes in cryptophytes. Here, a careful analysis of isolated antenna complexes from *Hemiselmis andersenii* show that a range of antenna complexes are expressed and that this cocktail can have a functional role, schematically conjectured in Fig 4. Part of the conjecture is that HaPE645 mediates energy transfer from the majority light harvesting complexes to the membrane-bound chlorophyll-containing complexes, which is a compelling idea. To do so successfully, the author posit that these proteins associate with the membrane preferentially--an intriguing idea. Such a motif is not unknown in photosynthesis, it is essentially the role played by the FMO complex in green sulfur bacteria.

Reviewer #3 (Remarks to the Author):

Cyanobacteria, red algae, and cyanophyten algae contain water-soluble, cytoplasmically located, pigmented phycobiliproteins (PBP), which form a variety of supramolecular complexes for light harvesting for photosynthesis. One additional group of algae, the cryptophytes (cryptomonads), also produce PBP, but in these organisms, the PBP are located in the thylakoid lumen. Cryptophytes are evolutionarily related to red algae and arose from a secondary symbiosis between a eukaryote that acquired a red algal symbiont. In recent years thanks to cryo-electron microscopy, remarkable progress has been achieved in structural studies of PBS, the light-harvesting complexes formed from PBPs in cyanobacteria, red algae, and cyanophytes. On the other hand, in spite of many years of study, the organization of PBP in cryptophytes remains enigmatic and unresolved.

The study of Rathbone et al. seeks to change this situation by systematically and comprehensively attacking this problem using a variety of methods, including transcriptomic analysis and inference, chromatography, mass spectrometry, spectroscopic methods, and X-ray crystallography. The results are impressive, as they identify many component proteins, some differing by as little as single amino

acids from others, and by obtaining X-ray structures for many of the proteins, including members of three spectroscopic classes. For the most part, this is an impressive study and while it does not solve the structural problem, it provides valuable new insights into how cryptophytan biliproteins are organized. There are a few minor issues, however, that the authors must address. Comments follow.

1. A major issue in this work concerns the issue of all whether all of the alpha subunit sub-variants are found in all cells of the culture or in members of subpopulations. The authors indicate that they serially subculture this organism (and don't even bother to maintain axenic cultures because they don't autoclave the medium). From the growth conditions described, which are clearly not going to produce exponential growth, they have created conditions that can (and almost certainly will) lead to mutant subpopulations within their cultures. These could have very similar but slightly different PBP, just as the authors describe. In other words, their cultures may simply be very far from clonal, and this seems likely to have been the case for a long time, perhaps years. While it seems likely that the major species of PBP (PE550, PE560, and PE645) would be present in all, some of the minor variants observed might only be present in subpopulations. There is no way to easily resolve this issue from the information provided. If the authors believe their cultures are clonal and that they do not contain quasi-stable subpopulations, they should provide evidence for this. If they maintain frozen stocks of a wild-type population that they know to be clonal, then they should state this. If their cultures are not clonal, they should clearly state that some of the heterogeneity observed could arise from genotypic drift in their cultures. Ultimately, the only way to distinguish this might be to perform single cell analyses such as genome sequencing or transcriptomics based on single cells. This is a very important issue because it certainly affects how one builds a model for the light-harvesting apparatus in these organisms. A caveat here is that some light-harvesting components do exhibit remarkable heterogeneity within a single cell—for example, LHC I and II components associated with PSI and PSII are known to be heterogeneous. However, that heterogeneity is more like that associated with major PBP types and not variants differing by a few amino acids as described here. Whatever the case, the authors must provide some information to address this issue.
2. A weak part of the argument for the model presented is the assertion that the PBPs form filaments in the thylakoid lumen. One presumes that the authors have selected their best TEM images to support this claim, and yet this reviewer was unable to unambiguously see filaments in the images provided. Perhaps the authors can be more explicit about where these filaments occur in the images or enhance the presentation.
3. While it is clear that the authors have obtained very high-resolution structures for the component proteins, they cavalierly state that they can distinguish phycocyanobilin from phycoerythrobilin, which are isomeric structures, and these from dihydrobiliverdin which differs from the other two tetrapyrroles by a single double bond. They show no evidence to support this assertion. While it may be the case that they can, they should provide some evidence that they can indeed distinguish the identities of the chromophores unambiguously, because the positioning of the chromophores and their identities are critical to understanding energy transfer pathways in these proteins.
4. If only one protein has been observed to form filaments, does this mean that other proteins, or the proteins in other cryptophytes, are differently organized? How general can this model be if it may only explain the organization for one protein in one organism? Wouldn't the authors expect that there are similar principles of PBP organization in most cryptophytes?
5. The authors suggest that PE645 may associate with another protein on the surface of the thylakoid lumen. Is there evidence that under some conditions PE645 is membrane associated? Are there conditions such that it is only released into solution by use of a detergent? Evidence of a membrane association would strongly support the model presented here.

Minor Points

1. Line 38. Are there red algal PBS as small as 1.2 MDa? The smallest PBS in red algae are about the same size as cyanobacterial hemidiscoidal PBS, which would be 4-5 MDa. *Rhodotella violaceus* has such small hemidiscoidal PBS. If there are smaller, please provide a reference.
2. Line 72. Insertions, not incursions
3. Line 76. Functional, not function

4. Line 106. What does "tightly peaked spectra" mean?
5. Line 118. Nearly identical, not near(-)identical
6. Line 125. Is "knob and hole packing" a common term? If so, provide a reference for it.
7. Line 161. Outnumbering, not out numbering
8. Line 320. Temperature range not temperaturerange
9. Line 333 and elsewhere. What is the purpose of a lysozyme treatment. Lysozyme is specific for linkages mainly found in peptidoglycan but somewhat for linkages in chitin, and to this reviewer's knowledge, neither occurs in cryptophytes.
10. Line 361. The abbreviation for Dalton is Da, not D
11. Line 411. Reversed-phase, not reverse phase
12. Line 529. "put down" ??
13. Lines 532 and 557. harbor or contain not host

Response to Reviewers' comments:

We have copied the complete reviewers' comments in blue. Our responses are in black print and always start on a new line: **Response:**

Reviewers' comments:

Reviewer #1 (Remarks to the Author):

This is an extremely impressive paper that describes the antenna system in a cryptophyte alga in remarkable detail. Beginning with sequence analysis, proceeding through protein isolation and identification, structure determination and finally the proposal of a detailed organizational model, the authors have produced a classic study that sets a standard for such work. I enthusiastically recommend that it be published after minor revision.

My main complaint centers around the proposal of a putative tether for the PE645 complex with the photosystem. I think it is reasonable to propose such a tether, as the authors show with the FRET calculations. However, there is not a consistent term used to describe it. In various places it is described as an adaptor, scaffold or tether.

Response:

We have checked both the MS and the SI and we believe we have been consistent in our usage of all of the terms: adaptor, scaffold or tether. *HaPE645* is almost exclusively referred to as “adaptor” (6 times in the MS and 7 times in SI). The word “scaffold” is only used once in the MS where it is used as a verb to describe the *LI* loop holding a chromophore in the open-braced form (not *HaPE645*). The word “tether” is used as a verb to hold *HaPE654* to the membrane (once in the MS and twice in SI). We believe that there should be no confusion.

Additionally, we have used the construct “terminal acceptor” (not mentioned by Reviewer #1) twice in reference to *HaPE645* (once in MS and once in SI). In both instances it states that *HaPE645* is the “...terminal acceptor of the soluble...antenna and adaptor to ...the integral membrane...systems”. Thus, we clearly differentiate these two functions of *HaPE645* – “terminal acceptor” and “adaptor”.

The authors should choose one word and be consistent, and also be up front about the fact that there is no direct evidence for it, so it is somewhat speculative.

Response:

Page 6, line 43: We have altered the start of the paragraph suggesting *HaPE645* is associated with the membrane. We now explicitly state that this is a speculation.

The paragraph used to start: “This implies that under normal physiological conditions *HaPE645* would be the closest protein to the membrane...”

It now starts: “Based on this evidence, we speculate that under normal physiological conditions *HaPE645* should be the closest protein to the membrane...”

I am not aware of any time-resolved studies on whole cells of this alga. If there are any, they should be cited, and if not, this provides an ideal way to test the organizational model. Such studies are outside the scope of this paper, as it is already very extensive, but perhaps should be mentioned as a way to test the model.

Response:

We agree with Reviewer #1. There are unfortunately no whole cell studies of any cryptophyte alga and such studies are outside the scope of this paper.

Reviewer #2 (Remarks to the Author):

This is an insightful paper that I recommend for publication in its present form. It has long been a mystery why there are so many "extra" alpha subunit genes in cryptophytes. Here, a careful analysis of isolated antenna complexes from *Hemiselmis andersenii* show that a range of antenna complexes are expressed and that this cocktail can have a functional role, schematically conjectured in Fig 4. Part of the conjecture is that *HaPE645* mediates energy transfer from the majority light harvesting complexes to the membrane-bound chlorophyll-containing complexes, which is a compelling idea. To do so successfully, the author posit that these proteins associate with the membrane preferentially--an intriguing idea. Such a motif is not unknown in photosynthesis, it is essentially the role played by the FMO complex in green sulfur bacteria.

Response:

Reviewer #2 asks no questions. We appreciate their support of our work.

Reviewer #3 (Remarks to the Author):

Cyanobacteria, red algae, and cyanophytan algae contain water-soluble, cytoplasmically located, pigmented phycobiliproteins (PBP), which form a variety of supramolecular complexes for light harvesting for photosynthesis. One additional group of algae, the cryptophytes (cryptomonads), also produce PBP, but in these organisms, the PBP are located in the thylakoid lumen. Cryptophytes are evolutionarily related to red algae and arose from a secondary symbiosis between a eukaryote that acquired a red algal symbiont. In recent years thanks to cryo-electron microscopy, remarkable progress has been achieved in structural studies of PBS, the light-harvesting complexes formed from PBPs in cyanobacteria, red

algae, and cyanophytes. On the other hand, in spite of many years of study, the organization of PBP in cryptophytes remains enigmatic and unresolved.

The study of Rathbone et al. seeks to change this situation by systematically and comprehensively attacking this problem using a variety of methods, including transcriptomic analysis and inference, chromatography, mass spectrometry, spectroscopic methods, and X-ray crystallography. The results are impressive, as they identify many component proteins, some differing by as little as single amino acids from others, and by obtaining X-ray structures for many of the proteins, including members of three spectroscopic classes. For the most part, this is an impressive study and while it does not solve the structural problem, it provides valuable new insights into how cryptophytan biliproteins are organized. There are a few minor issues, however, that the authors must address. Comments follow.

1. A major issue in this work concerns the issue of all whether all of the alpha subunit sub-variants are found in all cells of the culture or in members of subpopulations. The authors indicate that they serially subculture this organism (and don't even bother to maintain axenic cultures because they don't autoclave the medium). From the growth conditions described, which are clearly not going to produce exponential growth, they have created conditions that can (and almost certainly will) lead to mutant subpopulations within their cultures. These could have very similar but slightly different PBP, just as the authors describe. In other words, their cultures may simply be very far from clonal, and this seems likely to have been the case for a long time, perhaps years. While it seems likely that the major species of PBP (PE550, PE560, and PE645) would be present in all, some of the minor variants observed might only be present in subpopulations. There is no way to easily resolve this issue from the information provided. If the authors believe their cultures are clonal and that they do not contain quasi-stable subpopulations, they should provide evidence for this. If they maintain frozen stocks of a wild-type population that they know to be clonal, then they should state this. If their cultures are not clonal, they should clearly state that some of the heterogeneity observed could arise from genotypic drift in their cultures. Ultimately, the only way to distinguish this might be to perform single cell analyses such as genome sequencing or transcriptomics based on single cells. This is a very important issue because it certainly affects how one builds a model for the light-harvesting apparatus in these organisms. A caveat here is that some light-harvesting components do exhibit remarkable heterogeneity within a single cell—for example, LHC I and II components associated with PSI and PSII are known to be heterogeneous. However, that heterogeneity is more like that associated with major PBP types and not variants differing by a few amino acids as described here. Whatever the case, the authors must provide some information to address this issue.

Response:

Cryopreservation of cryptophytes is a rather tricky and needs to be optimized for each species. Most groups working on cryptophytes maintain their stocks by serial subculturing because it keeps them quickly accessible and avoids the 6 weeks or so to recover a frozen culture.

Our culture was obtained from the National Center for Marine Algae and Microbiota (NCMA; Page 8, lines 36-38) where their stock is also maintained by serial subculturing. We have used strain CCMP644 which is derived from one of four cultures in NCMA (formerly CCMP) which were united by Lane and Archibald (2008; Reference 25 in the revised MS) in the new species *H. andersenii*, primarily using molecular characteristics (nucleomorph and nuclear SS18S rDNA) and provenance (Gulf of Mexico). Each of these four cultures was originally established from field material and deposited by Luigi Provasoli in the early 1980's. None of them are axenic.

We were asked if we used single cell isolates. Using a single cell to propagate a culture risks the danger of accidentally selecting a minor variant that is not representative of the population as a whole. Culture collections transfer a large enough inoculum to avoid the problem. Even after being subcultured for 40 years, these four isolates were still close enough in nuclear and nucleomorph 18S sequences to justify being put in the same species!

Furthermore, the transcriptomes of all 4 strains were sequenced as part of the MMETSP project (<https://www.imicrobe.us/#/projects/104>). They all contain numerous "open" and "closed" alpha subunit sequences with a near complete overlap, dominated by transcripts encoding the dominant "open" form of PE555. All four have near-identical homologs of *HaPE645* and *HaPE560* alphas. We state the fact that our transcripts were derived and reconstructed from four separate strains in our Methods.

As a final note: the first and only cryptophyte nuclear genome sequence (of *Guillardia theta*) shows 20 different PBP alpha genes. As discussed in our manuscript, there is strong evidence that all *G theta* PBP alphas are expressed as protein. Thus, the complex, multi-alpha subunit antenna structure is likely to be common to cryptophytes, in general.

Reviewer #3 also states:

From the growth conditions described, which are clearly not going to produce exponential growth,

Response:

This statement by Reviewer #3 is incorrect. We have traced the growth rate of our cultures by two methods (optical density and cell scoring) where the data fit to a logistic function with an initial exponential growth phase followed by a plateau at around 2 weeks (see Rathbone's PhD thesis <https://doi.org/10.26190/unsworks/1605>).

2. A weak part of the argument for the model presented is the assertion that the PBPs form filaments in the thylakoid lumen. One presumes that the authors have selected their best TEM images to support this claim, and yet this reviewer was unable to unambiguously see filaments in the images provided. Perhaps the authors can be more explicit about where these filaments occur in the images or enhance the presentation.

Response:

First, the TEM images are thin sections from freeze-substituted whole cells stained using heavy atoms. The resolution of such images is far too low to visualise any filaments in the thylakoid lumen. Nowhere in the MS or SI do we claim to see filaments in the TEM images. Instead, the evidence for filaments comes from the fact that *HaPE555* repeatedly forms filaments in five different crystal structures. This is clearly described in the first paragraph on page 4 of the MS and SI page 3 Supplementary Note 3.

In all places where we mention filaments, we have clearly flagged their potential biological significance as speculative. In the main MS, Page 7, lines 7-9, we state: “It is also possible that the rafts of ...filaments...antenna structure.” In SI Supplementary Note 3 – *HaPE555* filaments, paragraph 2, first sentence: “...it remains unclear if these filaments are biologically relevant, however...”. Finally, in SI page 10, Supplementary Note 8 – Model including filaments, we discuss the possible advantages of a filament structure, however, we note that we have clearly flagged this as speculation: the second sentence in SI Supplementary Note 8 states: “It is possible that filament formation occurs in the biological antenna...”. Thus, while attractive for many reasons as described in both the MS and SI, *HaPE555* filaments remain a speculative concept until proven by direct evidence in cells.

Additionally, see our response to point 4 below.

3. While it is clear that the authors have obtained very high-resolution structures for the component proteins, they cavalierly state that they can distinguish phycoyanobilin from phycoerythrobilin, which are isomeric structures, and these from dihydrobiliverdin which differs from the other two tetrapyrroles by a single double bond. They show no evidence to support this assertion. While it may be the case that they can, they should provide some evidence that they can indeed distinguish the identities of the chromophores unambiguously, because the positioning of the chromophores and their identities are critical to understanding energy transfer pathways in these proteins.

Response:

There are two key structural features that allow us to unambiguously differentiate the chromophores PEB, DBV and PCB. These are: (i) the planarity of the individual pyrrole rings (where the planarity of pyrrole ring A differentiates DBV – planar, versus PEB – non-planar); and (ii) the coplanarity of the bridging carbon linkage between pyrrole rings C and D (which is unique to PCB). At the resolution of our crystal structures, these two features are unambiguous and allow us to confidently identify the chromophores.

Page 15, line 45 to Page 16, line 27: To make this absolutely clear, we have introduced a new section into Methods: Identification of chromophores. This section is accompanied by a new SI Supplementary Figure 8. In this figure, we present crystallographic support for our chromophore assignment in the form of a polder OMIT map [Reference 47: Liebschner et al Acta Cryst D 2017]. Polder maps are improved OMIT maps where atoms modelling the chromophore are removed and the remaining structure refined to remove any phase bias or

phase memory. The calculated polder maps clearly show the nature of the bonding structure bridging between pyrrole rings C and D for the chromophores attached to Cys82 in the *HaPE645* structure. These maps unambiguously demonstrate that PCB is attached to the $\alpha\beta$ protomer while PEB is attached to $\alpha_L\beta$. The polder maps were already presented in Figure 3b and 3c in the main manuscript, however, the additional material makes this point absolutely clear.

4. If only one protein has been observed to form filaments, does this mean that other proteins, or the proteins in other cryptophytes, are differently organized? How general can this model be if it may only explain the organization for one protein in one organism? Wouldn't the authors expect that there are similar principles of PBP organization in most cryptophytes?

Response:

Page 4, lines 7-9: The first statement by Reviewer #3 is incorrect, as per the MS, in addition to *HaPE555* both *CpPE566* and *PsE545* closed form PBPs are arranged in filaments (references 10 and 25). Second, each of these three proteins (*HaPE555*, *CpPE566* and *PePE545*) constitute the major component of the soluble light harvesting antenna in their respective organisms.

Our rationale for proposing that the filaments may be biologically relevant is explained in SI page 3, Supplementary Note 3 (paragraph 2) and SI page 10 Supplementary Note 8. In addition, an organised structure, such as filaments, would naturally explain the restricted mobility of the PBPs as measured by FRAP (MS Page 6, lines 4-9 ; and Page 7, lines 5-9).

In addition, see our response to point 2 above.

5. The authors suggest that *PE645* may associate with another protein on the surface of the thylakoid lumen. Is there evidence that under some conditions *PE645* is membrane associated? Are there conditions such that it is only released into solution by use of a detergent? Evidence of a membrane association would strongly support the model presented here.

Response:

Page 6, line 43: We have no direct evidence that *HaPE645* binds to the thylakoid membrane. As per our response to a similar comment by Reviewer 1, we have altered the start of the paragraph suggesting *HaPE645* is associated with the membrane. We now explicitly state that this is a speculation.

The paragraph used to start: "This implies that under normal physiological conditions *HaPE645* would be the closest protein to the membrane..."

It now starts: "Based on this evidence, we speculate that under normal physiological conditions *HaPE645* should be the closest protein to the membrane..."

Minor Points

1. Line 38. Are there red algal PBS as small as 1.2 MDa? The smallest PBS in red algae are about the same size as cyanobacterial hemidiscoidal PBS, which would be 4-5 MDa. *Rhodotella violaceus* has such small hemidiscoidal PBS. If there are smaller, please provide a reference.

Response:

The 1.2MDa PBS is from *Acaryochloris marina*. Our reference for this statement is reference 5 (at the end of this sentence), the review by Adir *et al.* 2020. We could add the primary reference from this review (ref 61 in Adir 2020) but we believe it would be inappropriate as it is not relevant to our paper.

2. Line 72. Insertions, not incursions

Response:

Page 2, line 39: “incursions” replaced by “insertions”.

3. Line 76. Functional, not function

Response:

Page 2, line 41: “function” replaced by “functional”.

4. Line 106. What does “tightly peaked spectra” mean?

Response:

Page 3, line 29: we have replaced “tightly peaked spectra” by “unusually narrow spectra” and referenced Harrop *et al.* 2014, where it is shown that the spectral width of *HaPE555* is approximately half that of comparable cryptophyte spectra.

5. Line 118. Nearly identical, not near(-)identical

Response:

Page 3, line 41: “near” replaced by “nearly”.

6. Line 125. Is “knob and hole packing” a common term? If so, provide a reference for it.

Response:

Page 4, Line 3: We have replaced “knob and hole” by “knobs-into-holes”. The latter is a commonly used phrase to describe protein interface packing (145 papers in PubMed). The term is based on the original description of a helical coiled-coil,

Crick (1952) Nature **170**:882-3, however, this reference is unwarranted.

Finally, we have also replaced “knob and hole” with “knobs-within-holes” on Lines 4 and 6 of SI Supplementary Note 3.

7. Line 161. Outnumbering, not out numbering

Response:

Page 4, line 41: “out numbering” replaced by “outnumbering”.

8. Line 320. Temperature range not temperaturerange

Response:

Page 8, line 38: “temperaturerange” replaced by “temperature range”.

9. Line 333 and elsewhere. What is the purpose of a lysozyme treatment. Lysozyme is specific for linkages mainly found in peptidoglycan but somewhat for linkages in chitin, and to this reviewer’s knowledge, neither occurs in cryptophytes.

Response:

Lysozyme likely serves no purpose with regard to cryptophyte lysis, however, it is a component of a standard lysis cocktail that we use in our laboratory. Hence, we report what we have actually done.

10. Line 361. The abbreviation for Dalton is Da, not D

Response:

Page 9, line 34: “kD” replaced by “kDa”.

11. Line 411. Reversed-phase, not reverse phase

Response:

Page 10, line 37: “reverse phase” replaced by “reversed-phase”.

12. Line 529. “put down” ??

Response:

Page 13, line 15: “put down” replaced by “likely due”.

13. Lines 532 and 557. harbor or contain not host

Response:

Page 13, lines 18 and 43: “hosts” replaced by “contains”.

REVIEWERS' COMMENTS:

Reviewer #1 (Remarks to the Author):

The revised manuscript is acceptable for publication in its current form.

Reviewer #3 (Remarks to the Author):

The revised manuscript of Rathbone et al. and their responses to reviewer comments satisfy this reviewer. The added text and figure supporting the identification of the chromophores is compelling and certainly addresses the prior comment adequately.